# Binding moral values gain importance in the presence of close others

Daniel A. Yudkin[1✉], Ana P. Gantman[2], Wilhelm Hofmann[3] & Jordi Quoidbach[4]

A key function of morality is to regulate social behavior. Research suggests moral values may be divided into two types: binding values, which govern behavior in groups, and individualizing values, which promote personal rights and freedoms. Because people tend to mentally activate concepts in situations in which they may prove useful, the importance they afford moral values may vary according to whom they are with in the moment. In particular, because binding values help regulate communal behavior, people may afford these values more importance when in the presence of close (versus distant) others. Five studies test and support this hypothesis. First, we use a custom smartphone application to repeatedly record participants' ($n = 1166$) current social context and the importance they afforded moral values. Results show people rate moral values as more important when in the presence of close others, and this effect is stronger for binding than individualizing values—an effect that replicates in a large preregistered online sample ($n = 2016$). A lab study ($n = 390$) and two preregistered online experiments ($n = 580$ and $n = 752$) provide convergent evidence that people afford binding, but not individualizing, values more importance when in the real or imagined presence of close others. Our results suggest people selectively activate different moral values according to the demands of the situation, and show how the mere presence of others can affect moral thinking.

[1] Social and Behavioral Science Initiative, University of Pennsylvania, Philadelphia, PA, USA. [2] Department of Psychology, Brooklyn College (CUNY), New York, NY, USA. [3] Faculty of Psychology, Ruhr University Bochum, Bochum, Germany. [4] Department of People Management and Organisation, Universitat Ramon Llull, ESADE, Sant Cugat, Spain. ✉email: dyudkin@sas.upenn.edu

In the course of daily life, people are apt to have a variety of social encounters, each demanding a set of contextually appropriate behaviors. For example, playing with one's child is likely to entail a very different set of actions than asking one's boss for a raise. As a result, successfully navigating the social world depends on people's ability to flexibly adapt their behavior according to various relational needs[1]. One cognitive tool people may use to help them do this is a set of moral values. Morals are standards for social living that provide people with guidance on how to treat each other[2–4]. As such, they help direct and constrain behavior in the context of different relationships[5]. Supporting this view, recent empirical and theoretical evidence suggests morality plays a critical role in many aspects of social life, including promoting cooperation[6], helping people pursue shared goals[7], regulating social relationships[8], and fostering group cohesion[9].

Recent theoretical advances in moral psychology suggest that human values may be divided into two distinct sets[10,11], each serving a different function. The first set, called "binding" values, govern people's behavior in local groups and communities. Based on tenets of "coalitional psychology,"[12] these values help regulate behavior in cooperative settings[13] and include loyalty, which encourages the subordination of individual motivations to the collective[14]; authority, which promotes the respect of laws, leaders, and traditions[15]; and purity, which helps to protect people and communities against potentially harmful pathogens[16]. The other set consists of "individualizing" values. Built on the "ethic of autonomy,"[13] these are concerned with ensuring personal rights and freedoms[17] and include values of fairness (based on conceptions of justice as rooted in altruism and equitable exchange[18]) and care (based on psychological imperatives to avoid doing harm to others[19]). These values serve to prosocial behavior outside of the context of group membership.

Conceptual schemas tend to become activated in contexts in which they are relevant[20,21]. For example, the act of sitting behind the wheel of a car is likely to make salient a variety of concepts related to traffic laws (stop at red) and vehicular operation (press clutch to change gears). This helps the mind efficiently store and process diverse swaths of information, since it would not be helpful to be rehearsing automotive facts while, for instance, attempting to prepare a pot of coffee[22]. In this way, the mental system allows for the efficient storage of information while remaining able to recruit it in situations in which it may prove useful[23]. Supporting this idea, research has shown that incidental cues in people's environment can activate mental representations that subsequently impact people's attitudes and behavior[24–26].

This fact, combined with the idea that moral values serve to regulate group behavior, leads to the possibility that people's moral values may be differently activated merely according to whom they are with in the moment. Existing work supports the idea that moral thinking may be impacted by subtle cues in people's environment. Research has shown, for example, that reading the Ten Commandments or writing a story in morally laden terms can strengthen people's motivation to act morally[27], and other research has shown that activating different emotion concepts can change prosocial behavior toward vulnerable social targets[28]. Past research also demonstrates the power of social context to impact moral behavior, including increasing egalitarianism[29], moralistic punishment[30], and cooperation[31]. More generally, classic research in psychology has shown how the "mere presence" of others can cause meaningful shifts in people's attitudes and behavior[32,33]. This suggests that the presence of others may affect the importance that people afford moral values.

If relational context does indeed influence moral thinking, then close others may have a particularly pronounced effect. Research shows that people feel heightened moral emotions toward close versus distant others[34], with corresponding impacts on evaluations and behavior[35], including increased empathy[36], generosity[37], leniency[38], and respect[39]; other work shows that people are more likely to cooperate with in-group (versus out-group) members[40]. Furthermore, participation in group activities can inspire prosocial emotions, including a sense of loyalty[14] and solidarity[41], and close relationships foster an increased sense of moral obligation[42]. And research has shown that increasing the accessibility of close others affects the way people construe and experience goals[43]. Thus close others may exert an especially strong effect on the importance people afford moral values.

Individualizing values apply universally to all people, while binding values tend to govern behavior in communities[44,45]. Accordingly, the effect of close others on the importance people afford moral values (hereafter termed "moral importance") may be more pronounced for binding than for individualizing values. Corroborating this view, research has shown that binding and individualizing values respond differently to different psychological cues. For example, abstract thinking, which tends to emphasize objects' universal properties[46], increases people's preference for individualizing versus binding values[47], suggesting that the latter may be more "context sensitive"—that is, responsive to variations in people's immediate social surroundings.

Overall, the reasoning outlined above leads to a set of predictions regarding the impact of social context on moral importance: specifically, that being in the presence of close others will increase the importance people afford binding, but not necessarily individualizing, foundations.

Testing how social context influences moral evaluation is important for several reasons. First, scholars have long considered morality the "social glue" that enables human cooperation. Understanding how moral values are shaped by context may help shed light on the cognitive processes that support this phenomenon. Second, this research may illuminate why people act in ways that (seemingly hypocritically) countermand their stated beliefs[48]. Consider a devoted husband who helps his wife destroy evidence of a crime. While, in the abstract, he might consider such behavior a violation of impartial justice, in the presence of his wife, loyalty might supersede justice[49]. While past research has proposed several possible explanations for this tendency[48,49], our research has the potential to introduce a new one—namely, that certain moral values become "activated" in specific contexts, leading people to behave in one circumstance in a way they may not in another. Understanding how moral concerns fluctuate according to the social exigencies of the situation may thus provide a new conceptual lens through which to understand apparent inconsistencies in people's moral behavior.

We conducted six studies to test how social context affects moral importance. Study 1 used an app-based experience sampling approach to solicit from participants carrying out their daily lives their views on the importance of various moral behaviors and details on whom (if anyone) they were with. Study 2 consisted of a large-scale preregistered online survey with a novel set of moral measures designed to provide convergent validity to the initial findings and to obtain a more precise estimate of observed effects. Study 3 was a laboratory-based experiment that randomly assigned participants to either an "alone" or a "partner" condition, measured how close they felt to their partner, then assessed their moral importance via the Moral Foundations Questionnaire[10]. Study 4A consisted of a preregistered experiment in which we attempted to manipulate moral importance via a manipulation of social closeness. In Studies 4B and 4C, we successfully manipulated moral importance by randomly assigning people to imagine they were in the presence of a close versus distant other. Overall, we find evidence that people consider binding values more important when in the presence of close

others. All materials are available at https://osf.io/4q8jg/, all analyses were conducted in R 4.1, and all hierarchical models defined in lme4[50]. We report all primary and preregistered analyses, all data exclusions, and all conditions across all studies.

## Results

Study 1 used an app-based experience-sampling approach to investigate the association between people's social context and moral importance. We employed this approach because past research has shown that experience-sampling strategies can provide insights into moral thinking and behavior that could not be achieved through other methodologies[51–53]. We surveyed a large sample of individuals and asked them to indicate whom they were currently with, as well as the importance they afforded various moral values, as they went about their daily lives (see "Methods"). Then, using a separate sample, we collected estimates of the "social closeness" of each of these relationship categories. We then tested whether moral importance varied according to relational context.

Overall, we collected 13,123 evaluations of moral behavior at 1951 different timepoints. Of these, 1145 (58.7%) were collected in the presence of another person. The most frequent type of social context was being with a romantic partner ($n = 316$), a child ($n = 215$), or a colleague or classmate ($n = 138$). The least frequent contexts were clients ($n = 13$) and best friends ($n = 19$) (For a complete list of relational context frequencies, see SOM 1.3.). No participants were excluded from the analysis.

First, we tested whether overall moral importance depended on relational context by performing a mixed-model analysis of variance (ANOVA) with relationship type as the primary predictor and moral importance as the dependent variable, controlling for age, gender, time of day, and day of the week (incidental variables), and setting participant as a random intercept. Results showed a main effect of relationship type, $F(4, 1608) = 5.74$, $p < 0.001$, suggesting the moral importance varied according to relational context. This effect was moderated by an interaction with value type, $F(4, 2891) = 2.68$, $p = 0.030$, suggesting that the effect of social context on moral importance differed depending on whether people were rating binding or individualizing values. Simple effects tests showed that binding values varied more according to social context than did individualizing values, $F(4, 3553) = 10.56$, $p < 0.001$ versus $F(4, 3016) = 5.93$, $p < 0.001$.

To test how moral importance correlated with social closeness, we asked a separate online sample ("Sample 2", $n = 94$) to rate how close, on average, they felt to members of each the relationship categories indicated by participants in the original sample. We examined the association between the average moral importance elicited in the original sample by each social context and the average closeness scores obtained from Sample 2 for that relationship. The model was a hierarchical linear model controlling for all incidental variables, with closeness as the primary predictor and moral importance as the dependent variables, and with participant and relationship type set as random effects. Results showed a main effect of social closeness on moral importance, $B = 0.03$, SE = 0.01, $t(1750) = 4.38$, $p < 0.001$, suggesting that people rated moral values more important the closer they were to whomever they were with. To generate a simplified understanding of these results, we computed an estimated moral importance associated with each relationship by adding the model intercept to the corresponding beta weight for each relationship and plotted it against the social closeness of that relationship obtained from Sample 2. Results showed a significant correlation, $r(14) = 0.77$, $p = 0.001$ (Fig. 1A), suggesting a positive association between the rated social closeness of each relationship type and the overall moral importance given by people in that social context.

Finally, we sought to test whether the relationship between social closeness and moral importance varied according to value type (binding versus individualizing). To do this, we centered moral importance on the respective means of each value type, then ran a linear model with the value type × closeness interaction as the primary predictor and moral importance as the dependent variable. Results showed a significant interaction, $B = 0.027$, SE = 0.014, $t(3561) = 1.98$, $p = 0.048$. Specifically, while both value types were positively associated with social closeness, this effect appeared to be stronger for binding values ($B = 0.05$, SE = 0.01, $t(3534) = 4.59$, $p < 0.001$) than individualizing values ($B = 0.02$, SE = 0.01, $t(3531) = 2.00$, $p = 0.045$; see Fig. 1B). This effect, though small, was robust to the effects of people's current affective state (see SOM 1.1 and 1.2). Thus it appears that, while being in the presence of close others is associated with increases in the importance of both value types, this relationship is stronger for binding than individualizing values.

Study 2 consisted of a large preregistered online sample that would allow us to determine with greater precision the relationship between social closeness and moral importance. We recruited a total of 2016 participants and asked them to indicate whom, if anyone, they were with. We then asked them two sets of questions pertaining to the importance of moral values. The first was identical to those presented in Study 1 (termed the "moral behavior" questions). The second consisted of a set of "decontextualized" moral values (termed the "moral value" questions), which allowed us rule out the concern that the questions were merely about social behavior (see "Methods"). We also collected information on political orientation in order to help rule the possibility that this was associated with the observed effects. In addition, we collected two measures of social closeness: one from an external sample; the other from participants themselves. In this way we sought to show that the relationship between binding importance and social closeness held across measures of morality and for both internal and external ratings of social closeness. The study was preregistered at AsPredicted.org; we report all preregistered analyses below.

Overall, 32% of the sample ($N = 647$) indicated they were in the presence of others. Of these, the most frequent were romantic partner ($N = 313$), child (188), and best friend (114). The sample had a mean political orientation of 2.75 on a 1 (very liberal) to 5 (very conservative) scale, meaning that it leaned slightly liberal. Each of the "moral behavior" items were closely correlated with the "moral value" items (all $p$s < 0.001).

We sought to test the relationship between the moral importance of individualizing versus binding values and social closeness. There were two ways of doing this. The first was to examine the relationship between moral importance and the estimated closeness ratings obtained from the separate sample; the second to examine the relationship between moral importance and the self-rated closeness of each relationship. The advantage of the first method is that it echoes the analysis strategy from Study 1 and helps to rule out the possibility that any observed relationships are the result of a correlation between people's tendency to rate moral values as important and others as close. The advantage of the second is that it more closely reflects the relationship between moral importance and people's actual perceived closeness to the person they were with. We report the results of each of these analyses in turn.

Regression results showed that the relationship between estimated closeness and moral importance differed according to value type, as revealed by significant value type × closeness interactions both for moral behavior, $B = -0.12$, SE = 0.01, $t(3531) = -16.10$, $p < 0.001$, and moral values, $B = -0.08$, SE = 0.01, $t(3513) = -12.56$, $p < 0.001$. Closer inspection of the data revealed that the importance of binding ($B = 0.05$, SE = 0.01, $t$

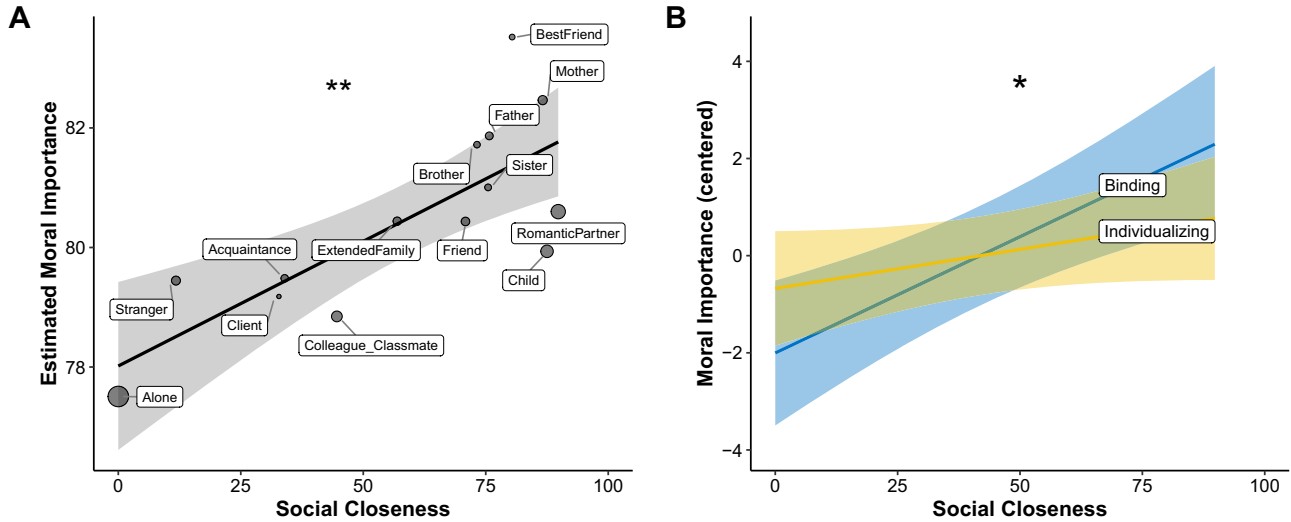

**Fig. 1 The relationship between moral importance and social closeness. A** The estimated importance participants ($n = 1166$) in each relational context afforded moral values plotted against the average social closeness of that relationship (obtained from a separate sample, $n = 94$). Black line shows the best fit line using ordinary least squares (OLS); ribbon = SEM; dot size reflects group size (range: 13—client to 806—alone). **B** The relationship between average social closeness and moral importance of binding and individualizing values, with both value sets centered on their respective means. Ribbons = SEM. **\*\*Slope significant, $B = 0.03$, SE = 0.01, $p < 0.001$; \*Difference of slopes significant, $B = 0.027$, SE = 0.014, $p = 0.048$. Source data are provided as a Source data file.

$(4287) = 6.84$, $p < 0.001$) but not individualizing ($B = −0.06$, SE $= 0.01$, $t(4285) = −7.74$, $p < 0.001$) behaviors was higher in the presence of close others, as was the importance of binding ($B = 0.04$, SE $= 0.01$, $t(4442) = 4.79$, $p < 0.001$) but not individualizing ($B = −0.04$, SE $= 0.01$, $t(4442) = −5.75$, $p < 0.001$) values.

We next examined the relationship between people's ratings of moral importance and self-rated closeness, controlling for age, gender, political orientation, and mood. Results showed robust interactions for both moral behavior ($B = −0.11$, SE $= 0.01$, $t(3530) = −16.58$, $p < 0.001$) and moral values ($B = −0.09$, SE $= 0.01$, $t(3512) = −14.72$, $p < 0.001$). Specifically, we found that the importance of binding behaviors ($B = 0.07$, SE $= 0.01$, $t(2729) = 8.05$, $p < 0.001$) and values ($B = 0.06$, SE $= 0.01$, $t(2559) = 6.94$, $p < 0.001$) was positively associated with social closeness, while individualizing behaviors ($B = −0.05$, SE $= 0.01$, $t(2729) = −5.36$, $p < 0.001$) and values ($B = −0.03$, SE $= 0.01$, $t(2559) = −3.66$, $p < 0.001$) were not. Thus we find across both "moral behavior" and "moral value" question sets and for both self-rated and estimated closeness that binding, but not individualizing, values increase in importance in the presence of close others (see Fig. 2). In accordance with the preregistration, we also examined the main effects of social context (being alone versus being with others) on the overall importance of moral behavior and values (i.e., collapsing across binding and individualizing values). Results showed no significant effect for moral behavior, $B = −0.52$, SE $= 0.68$, $t(1993) = −0.77$, $p = 0.44$, but a negative effect for moral values, $B = −1.89$, SE $= 0.58$, $t(2718) = −3.25$, $p = 0.001$. This appears to be driven by the negative relationship between social closeness and individualizing values, possible explanations for which are discussed below.

Next, we tested whether moral importance varied according to value type (binding versus individualizing) and relational context. To do this, we conducted a two-way ANOVA with relationship type, value type, and their interaction as the primary predictors and moral importance as the dependent variable, controlling for gender, age, political orientation, and mood (incidental variables). We repeated the analysis for both moral behavio r and moral values. The main effects of relationship type were marginally significant and not significant for moral behaviors and values,

respectively, $F(4, 4225) = 2.12$, $p = 0.076$ and $F(4, 3987) = 1.50$, $p = 0.199$—a fact that may be the result of opposite trends for binding and individualizing values. Indeed, significant relationship type × value type interactions emerged for both moral behavior, $F(4, 3510) = 94.4$, $p < 0.001$, and moral values $F(4, 3499) = 57.43$, $p < 0.001$, suggesting that the association between relationship type and moral importance differed according to value type.

Next, we tested the overall effect of social closeness (estimated and self-rated) on the overall importance of moral behaviors and values (i.e., collapsing across binding and individualizing values). For estimated closeness, results showed no significant effect on moral values, $B = −0.01$, SE $= 0.01$, $t(2718) = −1.44$, $p = 0.149$, and a negative effect on moral behavior, $B = −0.01$, SE $= 0.01$, $t(2713) = −1.83$, $p = 0.067$. On the other hand, for self-rated closeness, results showed a significant positive relationship between self-rated closeness for both moral behaviors, $B = 0.01$, SE $= 0.01$, $t(2713) = 1.99$, $p = 0.046$, and moral values, $B = 0.02$, SE $= 0.01$, $t(2718) = 3.27$, $p = 0.001$. These mixed results, which are likely due to variations in the relative impact of binding versus individualizing value sets, further support the idea that these different value types are best considered separately (i.e., as an interaction effect), as opposed to as a single category.

Study 3 consisted of a laboratory investigation that tested whether moral values changed in importance depending on manipulated social context. Participants were randomly assigned to an "alone" versus a "partner" condition, then their moral importance was assessed via the Moral Foundations Questionnaire[10], which, because it differed from the single-item questions utilized in the previous two studies, offered the opportunity to test the generalizability of these findings. As a secondary analysis, we also asked participants how close they felt to their partner, in order to investigate whether the effects were moderated by perceived partner closeness. We also assessed and controlled for political ideology and current affective state in order to address the possibility that these factors accounted for the observations. We then tested the data for the same pattern of effects as that observed in Studies 1 and 2.

There were no condition-based differences in the degree to which participants felt pressured to respond to the questions in a

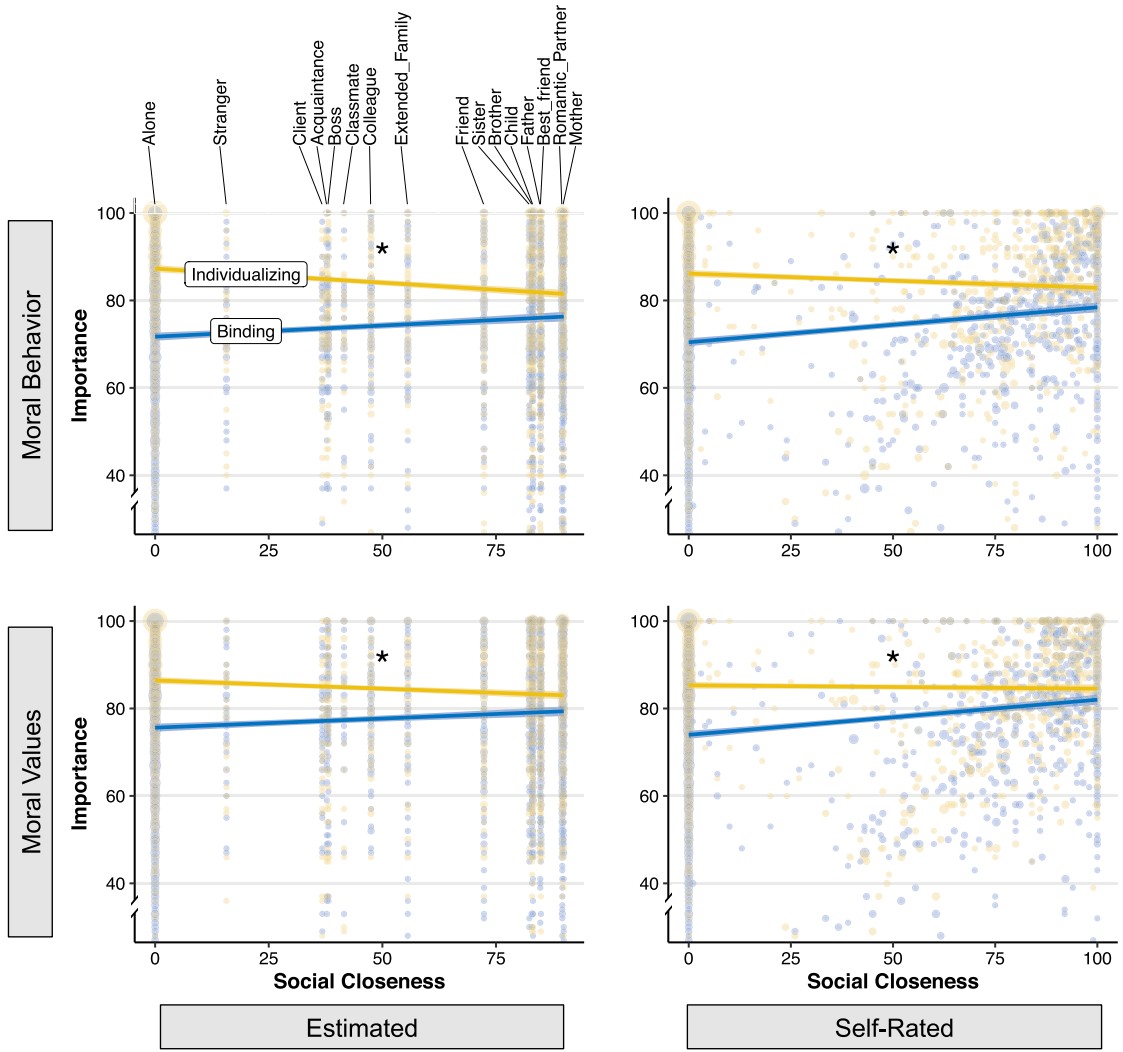

**Fig. 2 Social closeness predicts the importance people afford binding versus individualizing values.** The left two panels show the association between moral importance rated by people in the presence of various social relationships ($n = 2016$) and the social closeness of those relationships estimated by a separate sample ($n = 110$). The right two panels show the associations between moral importance and how close participants themselves felt to whomever they were currently with. The upper two panels reflect ratings of the importance of moral behaviors; the lower panels the importance of moral values. Alone participants were coded as having a social closeness of 0. Ribbons indicate SEM, and dots reflect individual ratings of moral importance, with sizes scaled to reflect multiple values on the same point. Points with moral importance <30 (<5% of total sample) have been omitted for the visualization. Source data are provided as a Source data file. *Differences between slopes significant at $p < 0.001$.

particular way, $t(388) = 1.11$, $p = 0.27$ or judged for their responses, $t(388) = 0.65$, $p = 0.52$, or in their current affective state, $t(375) = 1.32$, $p = 0.19$. The average political orientation was 2.3 on a 1 (very liberal) to 5 (very conservative) scale (SD = 0.96). The mean level of partner closeness was 2.11, SD = 0.98.

We first examined the effect of condition (alone versus partner) on overall moral importance, controlling for age, gender, mood, political affiliation, and non-moral evaluation ("incidental variables", see "Methods"). Results showed no significant effect, $B = 0.02$, SE = 0.06, $t(689) = 0.38$, $p = 0.701$. Next we tested the effect of condition on binding versus individualizing values. Once again, no significant effect emerged, $B = -0.09$, SE = 0.08, $t(346) = -1.12$, $p = 0.262$, and there was no simple effect of condition on binding values, $B = 0.07$, SE = 0.06, $t(683) = 1.18$, $p = 0.238$. Thus this study did not yield robust evidence that being in the presence of another person versus being alone causes an increase in the importance of binding values.

Because we did not obtain the results we had predicted, we conducted a series of exploratory analyses to further understand

the phenomenon. We first examined the frequency distribution of the closeness measure. Results showed that only 59 participants (32%) felt even "somewhat close" to their partner, suggesting a failure of our manipulation of social closeness. Given that our predictions suggested only people who felt close to their interaction partner would show an increase in the importance of binding values, we reasoned that the low number of participants in the former category might explain the absence of the predicted effect.

To test this possibility, we re-analyzed the results with a focus on the people who felt close to their partner. We split the participants in the "partner" condition into "distant" partner ($n = 120$) and "close partner" ($n = 59$) categories according to whether they indicated they felt at least "somewhat close" to their partner and treated these as separate conditions. We then conducted a 3 (social context: alone versus distant partner versus close partner) × 2 (value type: binding versus individualizing) mixed-model ANOVA with moral importance as the dependent variable, controlling for all incidental variables. We note this approach

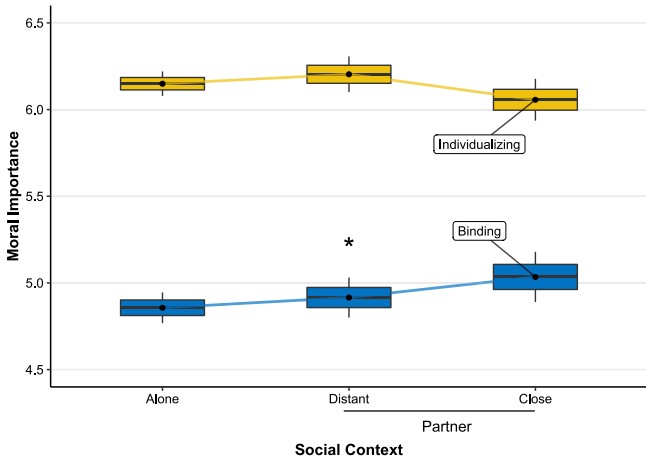

**Fig. 3 The association between social context (alone, distant partner, close partner) and moral importance according to value type.** Binding values were rated more important when participants were with a partner they rated as close than when they were alone. Box centers reflect group means; edges SEM; whiskers 95% CI ($n = 390$). Source data are provided as a Source data file. *Difference between means significant, $F(2, 340) = 3.69$, $p = 0.026$.

cannot provide causal evidence as participants were categorized according to closeness ratings they provided. As seen in Fig. 3, the analysis revealed a significant interaction between value type and social context, $F(2, 345) = 4.02$, $p = 0.019$. Specifically, binding values (alone: $M = 4.86$, SD $= 0.65$; distant partner: $M = 4.92$, SD $= 0.64$; close partner: $M = 5.03$, SD $= 0.56$) varied significantly with social context, $F(2, 340) = 3.69$, $p = 0.026$, but individualizing values did not (alone: $M = 6.15$, SD $= 0.52$; distant partner: $M = 6.20$, SD $= 0.57$; close partner: $M = 6.06$, SD $= 0.46$), $F(2, 340) = 1.63$, $p = 0.197$. This interaction effect remained significant even when additionally controlling for the effects of social desirability (i.e., the degree to which participants said they felt pressured to respond in a certain way to the questions, and the degree to which they felt judged for their responses), $F(2, 345) = 4.02$, $p = 0.019$, suggesting that the effects are not due to the effects of experimenter demand. Furthermore, a planned comparison of the importance of binding values among participants who were alone versus those in the "close partner" condition showed a significant difference, $t(107) = 2.09$, $p = 0.039$; supporting the claim that people who were alone considered binding values less important than those who rated their partner as close.

The results of the three previous studies were correlational. To obtain causal evidence for our effect, in Study 4A we recruited participants who said they were in the presence of another person. We then subjected them to a closeness manipulation: those in the "close" condition were asked to write about ways in which they felt close to the person or people they were with; those in the "distant" condition were asked to write about ways in which they felt distant or disconnected from the person or people they were with. After the manipulation, participants responded to a manipulation check that asked them how close they felt to whomever they were with and finally responded to a series of questions regarding the importance they afforded various moral values. We reasoned that, by creating a sense of social closeness or distance to whomever participants were currently with, we would observe a corresponding shift in the importance of binding values.

The most frequent social contexts were romantic partner ($n = 809$) and child ($n = 423$). Results showed the manipulation of social closeness was successful, as those in the "close" condition ($M = 82.2$, SD $= 22.9$) indicated feeling significantly closer to

whomever they were with than those in the distant ($M = 72.6$, SD $= 26.3$), $t(1948) = 8.65$, $p < 0.001$. The primary hypothesis consisted of a linear regression with condition as the primary independent variable and binding importance as the dependent variable, controlling for mood. Results showed no significant impact of condition on binding moral importance, $B = -1.15$, SE $= 0.84$, $t(1938) = -1.37$, $p = 0.169$. Thus our preregistered hypothesis was not confirmed.

In exploratory analysis, we examined whether self-rated closeness was associated with individualizing versus binding importance, controlling for age, gender, and mood. Once again, we found a significant value type × closeness interaction, $B = -0.08$, SE $= 0.02$, $t(1930) = -5.00$, $p < 0.001$, suggesting that the relationship between closeness and moral importance depended on whether people were rating binding or individualizing foundations. Moreover, we observed that the association with closeness was stronger for binding value, $B = 0.08$, SE $= 0.02$, $t(1926) = 4.94$, $p < 0.001$, than it was for individualizing ones, $B = 0.04$, SE $= 0.01$, $t(1925) = 3.48$, $p = 0.001$, thereby once again providing correlational, but not causal, support of our predictions (see SOM 4.1).

In Study 4B, we developed an alternative manipulation of social distance to provide causal evidence for the effect of social closeness on binding versus individualizing values. To do this, we created an exercise that prompted participants to imagine they were in the presence of a close versus distant other, after which we solicited their views on moral importance on the same 0–100 scale that we used in Studies 1 and 2. Specifically, in the "distant" condition, we asked participants to imagine they were sitting on a bench next to a stranger, while in the "close" condition, we asked them to imagine sitting on a bench next to their romantic partner. We selected these targets because they received the lowest and highest ratings of social closeness in Study 1 (The relationship of "mother" was rated slightly closer than romantic partner in Study 2, but because this relationship carries general connotations with "respect", we opted for a relationship that does not stack the deck so much in favor of the hypothesis.). This imagination exercise has the potential advantage of not eliciting a strong affective reaction to the close versus distant conditions, thereby prohibiting mood from overcoming the effects, as well as holding all situational elements constant by manipulating only the identity of the imagined interaction partner.

As predicted, condition had no effect on mood, $t(548) = 1.03$, $p = 30$. As indicated in the preregistration, the primary hypothesis consisted of a linear regression with condition as the primary independent variable and binding importance as the dependent variable, controlling for mood. Results showed a significant impact of condition on binding moral importance, $B = 12.05$, SE $= 1.76$, $t(540) = 6.86$, $p < 0.001$, thus confirming our preregistered hypothesis. Next we examined whether condition had an effect on individualizing importance, controlling for mood. It did, $B = 3.75$, SE $= 1.58$, $t(542) = 2.36$, $p = 0.018$. Moreover, the effect on binding values was significantly stronger than this effect, interaction $B = -8.30$, SE $= 1.62$, $t(540) = -5.13$, $p < 0.001$, thus confirming our prediction that binding values are more sensitive to social closeness than individualizing.

One possible concern with this study is that it relies on specific targets (stranger and romantic partner) to obtain its effect. While we see no a priori reason to imagine that these particular targets would yield the observed results in a different way than any other close versus distant other, a more generalized test of the hypotheses that does not rely on these specifics would lend credibility and generalizability to the results. For this reason, we conducted a final experiment 4C to determine whether the mere imagined presence of any close other is sufficient to change the importance of binding values.

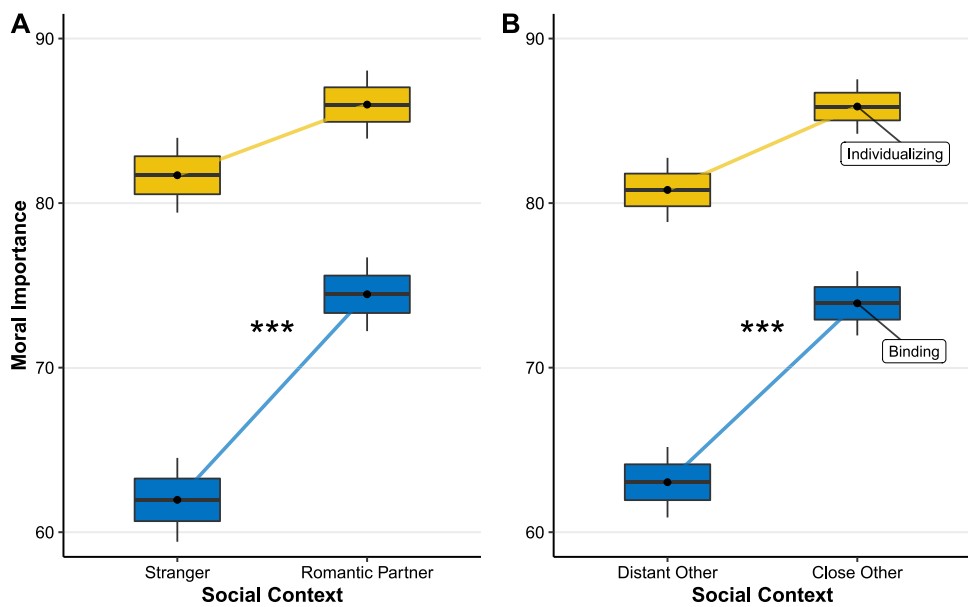

**Fig. 4 The effect of imagining being in different social contexts and the importance afforded moral values.** Each panel ($n = 580$ and $n = 752$, respectively) shows the average rated importance of binding versus individualizing values when people are in the imagined presence of another person. **A** Participants imagined being in the presence of a romantic partner or a stranger (Study 4B); **B** participants imagined being in the presence of someone whom they do or do not "feel close to" (Study 4C). Box centers reflect group means; edges SEM; whiskers 95% CI. Source data are provided as a Source data file. ***Difference between means significant, **A** $B = -8.30$, SE $= 1.62$, $t(540) = -5.13$, $p < 0.001$; **B** $B = -5.82$, SE $= 1.33$, $t(730) = -4.36$, $p < 0.001$.

Study 4C was identical to Study 4B with the exception that, instead of imagining sitting next to a "stranger" versus a "romantic partner", participants imagined sitting next to someone with whom they "feel close to" versus someone with whom they "don't feel close" to. As predicted, condition had no effect on mood, $t(734) = 1.26$, $p = 0.20$. As indicated in the preregistration, the primary hypothesis consisted of a linear regression with condition as the primary independent variable and binding importance as the dependent variable, controlling for mood. As predicted, we found a significant positive effect of an imagined close (versus distant) other on binding moral importance, $B = 11.08$, SE $= 1.47$, $t(727) = 7.55$, $p < 0.001$.

Next we examined whether condition had an effect on individualizing importance, controlling for mood. It did, $B = 5.15$, SE $= 1.30$, $t(730) = 3.96$, $p < 0.001$. However, the effect of binding values was significantly stronger, interaction $B = -5.82$, SE $= 1.33$, $t(730) = -4.36$, $p < 0.001$, thus confirming our prediction that binding values are more sensitive to social closeness than individualizing values (see Fig. 4).

## Discussion

Centuries of thought in moral philosophy suggest that the purpose of moral values is to regulate social behavior[2,54]. However, the psychology underlying this process remains underspecified. Here we show that the mere presence of close others increases the importance people afford binding moral values. By contrast, individualizing values are not reliably associated with relational context. In other words, people appear to selectively activate those moral values most relevant to their current social situation. This "moral activation" may play a functional role by helping people to abide by the relevant moral values in a given relational context and monitor adherence to those values in others.

Our results are consistent with the view that different values play different functional roles in social life. Past research contrasts the values that encourage cohesion in groups and relationships with those that emphasize individual rights and freedoms[10]. Because violations to individualizing values may be considered

wrong regardless of where and when they occur, the importance people ascribe to them may be unaffected by who they are with. By contrast, because binding values concern the moral duties conferred by specific social relationships, they may be particularly subject to social influence.

Our findings converge with work highlighting the practical contexts where binding values are pitted against individualizing ones. Research on the psychology of whistleblowing, for example, suggests that the decision over whether to report unethical behavior in one's own organization reflects a tradeoff between loyalty (to one's community) and fairness (to society in general)[55]. Other research has found that increasing or decreasing people's "psychological distance" from a situation affects the degree to which they apply binding versus individualizing principles[44]. For example, research shows that prompting people to take a detached (versus immersed) perspective on their own actions renders them more likely to apply impartial principles in punishing close others for moral transgressions[49]. By contrast, inducing feelings of empathy toward others (which could be construed as increasing feelings of psychological closeness) increases people's likelihood of showing favoritism toward them in violation of general fairness norms[56]. Our work highlights a psychological process that might help to explain these patterns of behavior: people are more prone to act according to binding values when they are with close others precisely because that relational context activates those values in the mind.

Our research is also broadly compatible with work on moral identity theory[41], which examines how people's moral self-concept impacts their moral judgments and decisions. For example, this work has found that situational cues which increase the mental accessibility of moral identity can subsequently increase prosocial behavior[24]. Our work builds on this by suggesting that one type of situation that may activate certain moral concepts is the presence of close others. Another finding from this literature is that people with highly self-important moral identity demonstrate a more expansive circle of moral regard toward outgroup members[57]. Building on this research, it would be fruitful

in future research to test whether moral self-identity moderates the effects we observed: perhaps people with such a self-concept are less susceptible to the influence of close others because they feel an extended sense of moral duty toward all humans.

This research adds to a growing body of work on how relational context impacts the activation and application of moral principles[5]. Other research in this area has shown, for instance, how judgments of different moral foundation violations vary across relational contexts[58], and how relationships may confer different moral obligations[59]. More generally, scholars have recently emphasized the importance of considering social context when investigating the psychology of moral judgment and evaluation[60,61]. The current research adds to this work by showing just how sensitive moral evaluation can be to how close people are to whomever they are with at the time.

Another implication of these results concerns the malleability and consistency of moral values. Psychologists often treat moral values as classic individual traits that consistently affect behavior across many situations, and that change only slowly over the course of a life span[10,62]. However, recent research on the contextual sensitivity of moral judgment and behavior has called this character-based approach to morality into question[63,64]. The current findings suggest one specific source giving rise to within-person variability or "inconsistency" in morality, by suggesting that, far from being immutable, certain moral values—in particular, binding values—may be flexibly applied according to the demands of the situation[65].

Readers will note negative relationship between individualizing values and social closeness in Study 2. This stands in contrast to the results of Study 1, which showed a positive relationship between these factors. This pattern is especially puzzling for the questions concerning moral behavior, which were worded almost identically to those of Study 1. One possible reason for this effect is that the true relationship between individualizing values and social closeness is close to 0, thus leading to apparent differences in the direction due merely to random sampling effects. However, the fact that the slopes differ significantly from zero suggests that there is more to the story. Another possible explanation for the difference concerns the composition and nature of the two samples (European smartphone users versus American Mechanical Turk users). Mechanical Turk workers, who typically complete survey on desktop computers, may be interacting with the survey in a different manner than members of the European smartphone sample. Determining how different cultures and forms of media engagement can affect the importance people afford moral values, while beyond the scope of the current project, will be an important topic for further research.

Study 4A found no evidence to support its preregistered hypothesis, which was that people who were prompted to write about ways in which they felt close to whomever they were with would rate binding values as more important relative to those prompted to write about ways they felt distant. One possible explanation for this is that people do not in fact rate binding values more important in the presence of close others, as we have hypothesized in this paper. However, there are two other possible explanations. The first concerns the effect of mood. Exploratory analyses suggested that condition had a strongly negative impact on mood, $t(1,939) = 5.61$, $p < 0.001$, doubtless a result of the fact that half of participants were prompted to feel distant from the people they were with (thereby leading them to feel negatively overall). Because mood is likely to vary much more readily than moral values, it is logical that the variations in mood would "drown out" any effect on moral importance.

Another possibility is that there are "compensatory" factors at play relating to the distance manipulation. When people are prompted to think about ways they feel distant or disconnected from the people they are with, they may simultaneously activate binding values such as loyalty and respect that encourage them to maintain social cohesion despite this distance. For example, someone who writes about ways their child annoys them may also be reminded of the importance of being loyal to one's child despite such relational perturbations. This might have the effect of increasing the importance of binding foundations in the distant condition as well, thus nullifying the effect. Another factor that may have exacerbated this phenomenon is the fact that the experiment was conducted during the coronavirus lockdown, a time in which people's proximity to close others would have rendered maintaining positive relationships particularly important for their psychological wellbeing. Studies 4B and 4C helped to answer these questions by using an imagined scenario that held all variables constant except the perceived closeness of the interaction partner.

Several new questions emerge on the basis of these findings. First, it will be important to determine how alternate mappings of the moral domain affect these results. For example, scholars have argued that four moral motives—unity, hierarchy, equality, and proportionality—may serve to regulate different relationships at different times[5]. Other research distinguishes between personal and impersonal forms of individualizing morality and includes such concepts as self-control and industriousness, or a more pluralistic conception of human morality, with ten or more discrete moral values[60]. Further research may fruitfully examine how different relational contexts impact the importance of moral values beyond the binding/individualizing dichotomy.

Future research could also investigate the way that specific values (e.g., fairness, loyalty, purity, etc.) interact with specific social contexts. For example, one might predict that values like respect (but not necessarily loyalty) would come into play when interacting with one's corporate boss. Analyses presented in SOM 3.1 and 4.2-3 show that different specific values interact with social distance in different ways across studies, suggesting that there is more to the picture here than a simple activation of all binding values uniformly. While our predictions concerned the broad impact of close others on binding versus individualizing values, it will be important in future work to specify more precisely which social conditions activate which particular values.

Next, the fact that we did not find the predicted effect in Study 4A raises some questions that could fruitfully be explored in subsequent work. We speculate that one possible reason why the closeness manipulation did not affect moral importance is because of compensatory processes that activate binding values in response to a relationship being under threat. Exploring this possibility, while beyond the scope of the current project, would be a promising subject of future research.

Finally, we note that, while three different questionnaires were used across the three studies in this paper, they all relied to some degree on explicit (i.e., self-reported) measures of moral importance. While the various study approaches used render unlikely the possibility that these effects would be the mere product of experimenter demand, future research will benefit from examining the extent to which these effects manifest via more implicit measures, including through reaction time or priming behavior. Understanding the degree to which social context can influence the implicit processing of moral information represents a promising domain of future work.

In sum, we find that what people value depends on whom they are with at a given moment. Specifically, people ascribe more importance to binding values when they are in the presence of close others, while the importance of individualizing values does not reliably change according to who is beside them. This is consistent with past theory suggesting that morality helps to regulate social relationships, and it suggests that different moral

values may be selectively activated according to the demands of the social situation.

## Methods

**Study 1, participants and method**. The research was part of the "58s" project investigating the everyday life of European adults[66], approved by The Ethics Committee of ESADE Business School, Spain, approval number 005/2019. Upon signing up for and providing consent to participate in the study, participants ($N$ = 1166, $M_{age}$ = 35.7, SD = 11.1, 861 females, 305 males) provided demographic information, including their age and gender. They were then contacted at random points over the next several months and asked to respond to a series of questions (pulled from a larger pool). No participants were excluded from the analysis.

Social context was assessed by asking participants whom they were currently with and then providing them a list of fourteen possible categories and asked to pick the one that best reflected their relationship to the person they were with (e.g., stranger, best friend, romantic partner). For analyses involving between-group ANOVAs, in order to avoid overly small group sizes (i.e., $n$ client = 13; best friend = 19) we categorized the relationships into five relationship types (alone, family, friends, professional, and other). Overall moral importance was assessed by averaging across responses on how important it was for participants in that moment to behave in accordance with each of five moral values[10] (care, fairness, loyalty, authority, and purity; 0—Not at all important; 100—Very important). The importance of "binding" values was assessed by averaging the loyalty, purity, and authority; the importance of "individualizing" values was assessed by averaging care and fairness. In addition, to assess current affective state, at approximately 75% of the timepoints, participants were also presented a question asking them to indicate how happy they felt (0—Not at all; 100—Very).

The analytic approach consisted of a series of multilevel analyses controlling for participants' age and gender, hour of day, and day of week, with participant-level random intercepts (incidental variables).

*Estimated closeness*. To an estimate of the average closeness of each of the relationships presented, we recruited a separate sample of participants on Amazon's Mechanical Turk ($N$ = 94, $M_{age}$ = 36.4, SD = 1.4, 54 female, 40 male) and presented them with a complete randomized list of the relationship categories we had provided to participants in the experience sample. We then provided the following instructions:

> Please use the slider to indicate how socially close that relationship is. It does not matter if that person exists in your own life; just indicate, for each relationship, how close that relationship is on average. (0—Not at all close to 100—Extremely close).

Mean social closeness scores were computed by averaging the closeness ratings for each relationship, with "alone" coded as 0.

*Materials (translated from the French)*. Social context: Who are you with?

- Alone
- Romantic partner
- Child
- Client
- Colleague or classmate
- Mother
- Father
- Friend
- Extended family
- Acquaintance
- Sister
- Brother
- Best friend
- Stranger

Moral values: (0—Not at all to 100—Completely):

- How important is it to you to act in a manner that is decent or pure?
- How important is it to you to avoid doing harm to others?
- How important is it to you to act in a manner that is honest and just?
- How important is it to you to act in a manner that is faithful and loyal?
- How important is it to you to respect hierarchy and authority?

**Study 2, participants**. The research was approved by the Institutional Review Board, University of Pennsylvania, Protocol #834222. The study was preregistered at https://aspredicted.org/blind.php?x=6sw3tn. Our power analysis was based on the overall main effect of the presence of others on moral importance in Study 1, with an approximate effect size of $d = 0.15$ (a very small effect). A power analysis conducted in G*Power suggested a required sample size of 1870 (power = 90%). To be conservative, we aimed to collect a sample of about 2000. No participants were excluded from the analysis, leaving a final sample of 2016 participants ($M_{age}$

= 36.9, SD = 11.8, 954 males, 1038 females, 24 other/fluid), who participated through Amazon's Mechanical Turk via TurkPrime[67] for compensation of 20¢. Questions for this and all subsequent surveys were administered through Qualtrics Versions 2019–2021.

*Procedure*. After providing consent, participants answered the same five moral importance questions that they responded to in Study 1, presented in random order (e.g., "How important is it to you to act in a manner that is faithful and loyal?"). We term these the "moral behavior" questions. Next, participants answered a series of five questions in the following format: "Please rate the overall importance of each of the following values" (0—Not at all important to 100—Extremely important) and then simply listed the value. The target values were "Fairness", "Loyalty", "Respect", "Purity/Sanctity", and "Care/Kindness". The purpose of this abstract question format was to ensure we had a measure of moral importance that was completely removed from any intrinsic social connotations and could thus be subject to concerns about the inherently social nature of the morality questions (We term these the "moral value" questions.).

Self-rated closeness: Next, participants indicated whether they were currently in the presence of other people and, if so, reported who by selecting any number of boxes corresponding to the different types of social relationships. We expanded the number of relationship types from 14 (in Study 1) to 17 (adding "boss", separating "colleague" and "classmate" into two separate responses, and adding "other") in order to provide more precision in our analyses (see "Materials"). Then, on the following screen, participants were asked "How close do you currently feel to your…" and each of the selections they had made on the previous screen were piped in so as to provide them an opportunity to indicate their self-rated closeness to each of the people in question.

Finally, participants reported their current affective state (0—Very negative to 100—Very positive), whether they felt pressured to answer in any particular way (1—Not at all to 5—Extremely), their age, gender, and their political orientation (1—Very liberal to 5—Very conservative).

The analysis consisted of a multilevel regression with participant set as a random factor in order to account for the fact that some participants were in the presence of multiple others at the same time, controlling for gender, age, political orientation, mood, and non-moral evaluation.

Estimated closeness: In order to obtain an additional estimate of the closeness of the various social contexts that people were in, we recruited a sample of 110 participants (60 males, 45 females, 5 other, $M_{age}$ = 36.2, SD = 11.4) through Amazon's Mechanical Turk and asked them to indicate how "close on average" was each relationship in the list of 17 presented to the primary sample, with the exception of "other" and "alone" (since these did not make sense in this context).

*Materials*. Social context: Are you currently in the company of other people? (Yes/No)

- [If yes]: Who?

    Romantic partner
    Child
    Client
    Colleague
    Classmate
    Mother
    Father
    Friend
    Extended family
    Acquaintance
    Sister
    Brother
    Best friend
    Stranger
    Boss
    Other

**Study 3, participants**. The research was approved by the Institutional Review Board, University of Pennsylvania, Protocol #834222. This study was implemented at the University of Pennsylvania with undergraduates as participants. The overall purpose of the study was to determine whether the importance people afforded moral values depended on whether they were with another person and how close they felt to that person. Thus, we planned to randomly assign participants to either the "alone" or the "partner" condition and then measure people's sense of social closeness to their partner. Because the social context that people were going to be in in the "partner" condition was that of "colleague/classmate", we based our power analysis off the analogous results in Study 1. Probing the difference in moral importance for people in this category (versus alone) yielded an effect size of about $d = 0.2$ (a small effect). Because the laboratory-based nature of the study allowed us to reduce additional variance introduced by other contextual differences (e.g., differences in the environment in which the questions were answered), we estimated an effect size of $d = 0.3$. A power analysis in G*Power suggested that a

minimum sample size of 278 is required to detect an effect of this size (power = 80%, one-tailed); we sought to oversample in order to maximize our changes of detecting an effect.

Overall, a total of 404 participants ended up registering to participate in the study, advertised as "Who Do You Think You Are?" via an online recruitment tool (SONA), in exchange for course credit. Of these, we excluded four participants for failing to follow instructions, four for failing an attention check, and six participants who knew experimenter beforehand, since a relationship between the participant and the experimenter introduced the possibility that a sense of being in a social situation would be activated even among participants who were in the "alone" condition. The resulting sample consisted of 390 participants ($M_{age}$ = 19.8, SD = 1.2, 267 females, 122 males, 1 other/fluid).

*Procedure.* Participants were randomly assigned beforehand to participate in the study either alone or with a partner by means of being given the opportunity to sign up for time slots with either one or two availabilities per slot. Participants were greeted at a waiting area by the experimenter, who informed them they would be answering some questions about their beliefs and opinions. The experimenter led them to an experiment room in which there were two computers on two opposing desks. The desks were arranged such that participants sitting at them would be facing each other but would be unable to view each other's responses. The purpose of this was to heighten the sense of being in the presence of another person while mitigating the possibility that people would feel that they were being judged for their responses.

The experimenter informed participants that they would begin by completing a "Getting To Know You" task. The intention of this task was to increase feelings of personal closeness through disclosure of personal information[68]. After providing consent, participants were handed a piece of paper with three question sets. Each question set consisted of four personal questions (e.g., "What are some of your hobbies?" and "If you could travel anywhere in the world, where would you go and why?"; see "Materials"). Participants in both conditions were instructed to complete the first two question sets, but not the third, which would be completed at the end of the study. The purpose of leaving the third question set unanswered was to promote the belief among participants in the "partner" condition that the social interaction had not yet concluded, thereby increasing people's sense that they were engaged in a social interaction. Participants in the "partner" condition were instructed to take turn asking each other the questions and write down their partner's responses on their own paper. Participants in the "alone" condition were instructed to provide their own responses to the questions and write them down on the paper.

After participants completed the first two question sets, they were instructed to notify the experimenter, who then instructed them to complete a set of questions on the computer screen. First, as a measure of social closeness, they indicated how close they felt to their partner (1—Not at all close to 5—Very close). Then they responded to the primary set of questions, which consisted of the 30 items comprising the "Moral Values Questionnaire" (MFQ). In accordance with past theory, the MFQ is designed to assess the importance that people ascribe to each of five moral values. The questionnaire also includes measures of non-moral evaluation (e.g., the importance of being good at math; the importance of having a good time) to be used as controls to ensure that moral importance did not just reflect people's tendency to rate all preferences (even non-moral ones) as important. In addition, participants provided a variety of demographic information, including age, gender, affective state (1—Not at all happy to 100—Very happy), and political affiliation (1—Very liberal to 5—Very conservative). In order to control for social desirability, participants indicated the degree to which they felt pressured to respond in a particular way to the questions (1—Not at all to 5—Very much so) and judged for their responses (1—Not at all to 5—Very much so). After completing the questions on the computer survey, participants were instructed to notify the experimenter, at which point they were debriefed and dismissed.

*Analytic approach.* The primary analysis consisted of a regression model with moral importance as the dependent variable, controlling for age, gender, affective state, the importance of the non-moral items, and political affiliation. The primary predictors were value type and social closeness, operationalized both as a manipulated factor (partner versus alone) and partner closeness (close versus distant). We predicted that people who felt close to their partner would rate binding values more important than those who were alone, and that this effect would differ from the effect of partner closeness on the importance of individualizing values.

*Materials.* Getting-to-Know-You Questions:
Question Set 1
1. What is your first name?
2. Where are you from?
3. What is a food you really enjoy?
4. What do you like the most about Philadelphia? What do you hate the most?
Question Set 2
1. What are some of your hobbies?
2. What would you like to do with your life in the next 5 years?

3. What is something you have always wanted to do but probably never will be able to do?
4. If you could travel anywhere in the world, where would you go and why?
Question Set 3

1. What is your happiest early childhood memory?
2. Describe the last time you felt lonely.
3. What is one recent accomplishment that you are proud of?
4. What is one of your biggest fears?
Moral Values Questionnaire: The wording for the MVQ can be found at https://moralvalues.org/questionnaires/.

**Study 4A, participants**. The research was approved by the Institutional Review Board, University of Pennsylvania, Protocol #834222. The study was preregistered at https://aspredicted.org/blind.php?x=fr9fd3. Our power analysis was based on pretesting that suggested an approximate effect size of $0.1 < d < 0.2$. A power analysis conducted in G*Power suggested that a required sample size of about 2000 (power = 90%) would be capable of detecting an effect size of $d = 0.14$. We only included for analysis participants who passed an attention check (how many fatal heart attacks they had had in their life), were not using a proxy server or VPN, and were in the presence of others. The final sample consisted of 2031 ($M_{age}$ = 32.2, SD = 11.7, 867 males, 1050 females, 26 other/fluid, 88 missing), who participated through Amazon's Mechanical Turk via TurkPrime[67] for compensation of 50¢.

*Procedure.* After consenting to participate, participants were asked who, if anyone, they were with. Next, they were randomly assigned to the "close" or the "distant" condition, in which they were prompted to write at least 50 characters describing ways in which they felt close to or distant from the people they were with. Next, participants responded to a manipulation check asking them how close they felt to the people they were currently with. Then they were asked how important were the following values (in randomized order): "Respect authority" (authority) "Be loyal" (loyalty), and "Behave decently" (purity); then "Avoid doing harm" (harm) and "Act fairly" (fairness). Next, they indicated their demographic information, mood, and were asked what they thought the study was about.

The preregistered analysis consisted of a linear regression with the averaged importance of the three binding foundations (purity, loyalty, and authority) as the dependent variable and condition as the independent variable, controlling for mood.

*Materials.* Social closeness manipulation: Participants in the close condition were provided the following instructions:

> Please describe some ways in which you feel **close** or **connected** to the person or people you're with. You may write about things you have in **common** with them, things you **love** about them, or a specific time in which you felt particularly **connected** to them. Remember your response is completely confidential and anonymous. You must spend at least 30 s on this page and write at least 50 characters.

Participants in the distant condition were provided the following instructions:

> Please describe some ways in which you feel **distant** or **disconnected** from the person or people you're with. You may write about ways in which you are **different** from them, things you **don't like** about them, or a specific time in which you felt particularly **disconnected** from them. Remember your response is completely confidential and anonymous. You must spend at least 30 s on this page and write at least 50 characters.

**Study 4B, participants**. The research was approved by the Human Research Protections Program at the Institutional Review Board, University of Pennsylvania, Protocol #834222. The study was preregistered at https://aspredicted.org/blind.php?x=f9796m. Our power analysis was based on pretesting that suggested an approximate effect size of $d = 0.3$. A power analysis conducted in G*Power suggested a required sample size of about 580 (power = 95%); we sought 600 to be conservative. We only included for analysis participants who passed an attention check (how many fatal heart attacks they had had in their life) and were not using a proxy server or VPN. The final sample consisted of 580 participants ($M_{age}$ = 32.5, SD = 11.5, 211 males, 367 females, 2 other/fluid), who participated through Prolific for compensation of 25¢.

*Procedure.* After consenting to participate and passing the attention check, participants were asked to imagine they were sitting on a bench next to a stranger or a romantic partner. They were then asked to indicate, in randomized order, how important all five values were to them (as in previous studies). They then provided demographic information, were probed for suspicion, and debriefed. Thirteen participants (2%) expressed some indication that they understood the hypothesis behind the study; all analyses stand when excluding these participants from analysis.

*Materials.* Distance manipulation: Imagine you are sitting on a bench next to a stranger. How important in that moment would each of the following moral values be to you?

Closeness manipulation: Imagine you are sitting on a bench next to your romantic partner. How important in that moment would each of the following moral values be to you?

**Study 4C, participants.** The research was approved by the Institutional Review Board, University of Pennsylvania, Protocol #834222. The study was preregistered at https://aspredicted.org/blind.php?x=f9796m. For this preregistered study, our power analysis was based on pretesting that suggested an approximate effect size of $d = 0.25$. A power analysis conducted in G*Power suggested a required sample size of about 800 (power = 95%). We only included for analysis participants who passed an attention check (how many fatal heart attacks they had had in their life) and were not using a proxy server or VPN. The final sample consisted of 752 participants ($M_{age} = 33.2$, SD = 12.5, 272 males, 479 females, 1 other/fluid), who participated through Prolific for compensation of 25¢.

*Procedure.* After consenting to participate and passing the attention check, participants were asked to imagine they were sitting on a bench next to someone they "feel close to" or "don't feel close to". They were then asked to indicate, in randomized order, how important all five values were to them (as in previous studies). They then provided demographic information, were probed for suspicion, and debriefed.

*Materials.* Distance manipulation: Imagine you are sitting on a bench next to someone you don't feel close to. How important in that moment would each of the following moral values be to you?

Closeness manipulation: Imagine you are sitting on a bench next to someone you feel close to. How important in that moment would each of the following moral values be to you?

**Reporting summary.** Further information on research design is available in the Nature Research Reporting Summary linked to this article.

## Data availability
The datasets generated for the current study are available in the Open Science Framework repository, https://doi.org/10.17605/OSF.IO/4Q8JG. Source data are provided with this paper.

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

## Acknowledgements
We thank Cliff Workman, Fiery Cushman, Geoff Goodwin, Hannah Read, Holly Engstrom, Ike Silver, Nina Strohminger, Kristopher Smith, and Yaacov Trope for their valuable feedback; Nykko Vitali, Sean Young, and Alina Mizrahi for help conducting research; Vani Kilakkathi for her support; and the Gilbert Lab, where many of the ideas for this project originated. Jordi Quoidbach thanks the Ministerio de Economía, Industria y Competitividad, Gobierno de España (Grant No. RYC-2016-21020) for financial support.

## Author contributions
D.A.Y. and J.Q. designed the research; D.A.Y. and J.Q. collected the data; D.A.Y. analyzed the data and wrote the paper, with critical edits from J.Q., W.H., and A.G.

## Competing interests
The authors declare no competing interests.
