## [Peer Review File · Nature Communications]

Reviewers' comments:

Reviewer #1 (Remarks to the Author):

Both methodologically (experience sampling) and substantively (the importance of morality in everyday life), this paper is interesting and important, and I warmly recommend publication (with pretty minor revisions). It's the kind of study we need more of.

Some comments follow, the most serious of which concerns some daylight between what is shown and what is sometimes claimed. I expect everything I mention is pretty readily addressable.

I should immediately note that I am trained as a philosopher, not a scientist, and am not qualified to assess statistical technique, and the like. So the editors should consult with someone more expert on those questions before going forward. I do note that the effects described are in the vicinity of "social priming," and given that such studies have been a special source of anxiety in recent discussion of replication, I'd have been helped by a bit of discussion of power, effect size etc. For example, Figure 1A makes the effect look pretty modest, a few point difference in "importance" on a 100 point scale. Is that a fair characterization? I realize that one can't do "metascience" all the time, if one wants to do some actual science, but I'd have been helped by some guidance here – and since NC is a general journal, and the paper is of general interest, other readers, hopefully some drawn from beyond social science, may feel the same.

ABSTRACT: There is a question about how findings are framed. From the abstract, one gets the impression that the hypothesis whether the "purpose" of morality is social regulation, a question we are told has not been "directly tested" before. I don't think the current study addresses this (indeed, the "purpose" issue might be difficult to assess empirically). The main finding, I gather, is an association between sociality and the importance assigned to values, which I'm happy to grant intimates a causal relation. But I don't see how this gets us to "purpose." Suppose we find that cloudy days are associated with people assigning more importance to moral values; should we then conclude the purpose of morality is cloudy days? (And the same for mood, and so on.) So I'd be inclined to make sure what's been shown is stated precisely – and I think more modestly. Even the "corroboration" language at the end of the abstract seems to me a reach.

p. 3: "the emerging view of moral values likens them to rules." I'm a rules guy myself, but this is contentious. In philosophy, "particularists" and virtue theorists dispute the importance of general rules in moral thought. Perhaps more seriously, psychologists like Cushman and Crockett favor "valuational" approaches. See recent work by Nichols and colleagues on the debate, which would be a good way to flag the lack of consensus. This is not a main issue for the paper, so it can be addressed at no cost.

p. 3. I'm also a fan of watching eyes, but I believe there our grounds for some reservation in the literature. This might be flagged.

p. 4. Last sentence of first para, “support.” This sentence also strikes me as outrunning what’s been shown. The next paragraph begins with a more careful statement, which I take to be the “official” finding.

p. 5. I more usually see the MFT cashed in terms of binding/individualizing rather than communal/universal. I think that’s what Graham favors, and I guess he should know. Universal raises complex issues, and is a tricky contrast with communal, since one might treat community as a universal value. Also, I don’t see why one is more important for social regulation than the other, a priori: is fairness less important for social regulation than purity or loyalty? (This does not detract from the paper’s relevant finding, which is cool.)

p. 5. Big gender skew for the sample. No gender differences?

p. 6. I wasn’t completely clear on the “positivity effect.” Bit more discussion of what was found? My fault no doubt, but not clear on contributions of positivity and sociality.

p. 8. First paragraph of the discussion also seems to me to encourage the entangling of the demonstrated association with the speculative purpose I noted with respect to the abstract.

p. 8. On the novelty of the result. It certainly seems to me novel in terms of directly addressing the paper’s official hypothesis (sociality > importance). But it might be (very) worth noting that the relation between sociality and morality has been the object of extensive study: there’s the watching eyes study that does get mentioned here, and things like the venerable research on the bystander effect, and variations on Milgram (eg, “defiant peer”), etc. A general readership would appreciate being alerted to some of this material – among the most important in social psychology.

p. 9. These qualifications are interesting and important. I think it would be worth further underscoring the uncertainties of self-report (with particular reference to experience sampling?) which may be more acute in the social contexts at issue.

P. 9. “called this trait-based morality into question.” Here’s another place where controversy ought be noted. Indeed, the controversy was one of the main issues that helped empirically informed, interdisciplinary moral psychology take off, starting around 2000 or so. Relatedly, the choice of citations for this struck me as unusual, in particular by not including any of the classic, widely cited, discussions that are central to the debate, such as Harman, Doris, and Ross and Nisbett. Again, general readers wishing to oriented with respect to this issue would benefit from being pointed to these treatments.

p. 10. “morality is social” I’m repeating myself here, but this does not follow directly from people assigning more importance to moral values in social settings.

Reviewer #2 (Remarks to the Author):

In this paper, the authors report a study using a novel, smartphone-based methodology and a follow-up lab study, which show that moral values become more important when one is in the presence of others than when one is alone. This seems to be especially true for the values of authority, loyalty, and purity.

This is an interesting finding, and the unique, real-world methodology deserves praise. However, I do have several concerns about the paper.

MAJOR CONCERNS

I am worried about the analytic strategy employed in the smartphone study. The authors control for age, gender, and time effects, but they do not account for individual differences that are known to be correlated with endorsement of moral foundations, particularly political views (see Graham, Haidt, & Nosek, 2009; Haidt & Graham, 2007; Landy 2016; dozens of other papers). This study is not experimental, so it could be that conservatives just tend to spend more time with people of close social proximity. This would not explain the overall effect of social proximity on value importance, but it could explain the difference in slopes between "communal" and "universal" values. If conservatives spend more time with family and friends than liberals, and they value "communal" morality more than liberals, that could produce the observed effect.

There are a few caveats to this point. First, I don't know if the authors have access to information about their participants' political views. If they don't, then this is a major limitation to the study that should at least be acknowledged. Second, the study was conducted in Europe, where the political landscape is somewhat different than in the US. But, while the "liberal/conservative" distinction means something a bit different across the pond, my understanding is that the "left/right" distinction is quite similar to the US. Third, I do not know of any research that suggests that conservatives spend more time with socially proximate others, but, on the other hand, I don't know of any research suggesting the opposite, either. So, this is an open question, and a live alternative explanation for this result. Notably, the lab study does *not* address this problem, because the two types of values are not analyzed separately. The authors could add this analysis, which would deal with this issue to some degree, but since the focus of the paper is very much on the smartphone study, I think this needs to at least be addressed rhetorically in that study as well.

Another issue is that there is a bit of a mismatch between theory and empirics, I think. On my reading, when moral psychologists say that morality is inherently social, they mean that morality is for accomplishing social goals (e.g., Haidt & Kesebir, 2010: "Moral systems... work together to suppress or regulate selfishness and make cooperative social life possible"). One might derive the prediction that moral values take on more importance in the presence of others from this view, but the authors make it seem like this is *the* core assumption of this view (e.g., the first sentence of the Discussion). It really

isn't. The core assumption is that morality serves a functional purpose. Gaining importance in social settings might help to facilitate serving that purpose, but it doesn't seem to be strictly necessary, nor is this part and parcel with existing theories.

MINOR CONCERNS

-p.6: "Conducting the same analysis without controlling for mood increased the strength of the effect." Yes, but only nominally. The difference here is between $\beta = .089$ and $\beta = .092$. Two effect sizes do not get much more similar than that. This is overstated.

-I find it odd that the authors use the terms "universal" and "communal". To harmonize with prior research, I would recommend using the usual terms "individualizing" and "binding".

-It would be helpful if the authors reported descriptive statistics for each condition in the lab study. With only the regression coefficients presented, it is a bit difficult to interpret what is happening.

Reviewer #3 (Remarks to the Author):

The authors present an interesting if early finding, that the scores on their "moral values" scale are significantly higher when measured in the presence of others than when not. This wrinkle in their data, perhaps predicted, is intriguing and warrants further inquiry, but comes well short of the scope of conclusions that the authors put forth, who present themselves as the first authors to "test directly" whether "the purpose of morality is to regulate social behavior" (see lines 1-3 of abstract). This claim is, to say the least, unwarranted, at multiple levels.

What the authors have is the finding that a bunch of French and Belgian smartphone users (3/4 women) rate a 5-item morality questionnaire higher on their smartphone when they also report being around other people. The "moral values" questionnaire comprises 5 items such as "How important is it to you to act in a manner that is faithful and loyal?" In other words, it blurs measures of moral motivation, of the self-importance of moral identity a la Aquino and Reed, of as the authors also call it, "moral values" or "moral evaluation." So there is not much to be gained theoretically about the measure of moral self-importance from this measure or this implementation. To give an example of where the lack of precision in the concepts really starts to make trouble, please see p. 9, "our work shows that the presence of others impacts not just moral behavior, but also moral evaluations, thus providing evidence of the later." There is not a iota of moral behavior or moral evaluation in the authors' measures. Moral evaluation would be typically understood as closer to moral judgment, as in when you evaluate a moral action, not the importance one attaches to a moral concern, which again is best understood as its relevance in given a social context, or at best a stable measure of self-importance a la Aquino & Reid. ("How important is it for you to wash your hands?" when preparing food illustrates how, in context and in everyday parlance, "how important" typically means how important is this concern in this context -- not a global evaluation -- which is quite likely how participants understood it)

The one tradition the authors tip their hat to is Graham and Haidt's five foundations, though I believe they rename Binding/Individualizing into Communal/Universal, presumably because that fits better the current model. This renaming does not seem necessary or warranted, unless the authors claim to be talking about something else. However, the authors do not present any data suggesting that their five items indeed tap into two different dimensions, and it seems like a lot to want to treat them as two distinct theoretical constructs. This is especially the case because some of the idiosyncrasies of the items awkwardly match some of the most salient categories: "hierarchy and authority" would be rated the highest in front of a parent, or high-status acquaintance or colleague, rarely alone. "Faithfulness and authority" are also more likely to be primed in the presence of a romantic partner and best friend. So the very words chosen, and the fact that they are most relevant in the context of specific relationship, would contribute to higher ratings in social settings, but making it difficult to want to draw big conceptual dimensions out of these 5 specific items. One analysis would be to break it down by type of item and type of other (e.g., if loyalty is primed with best friend, parent, romantic other), but short of these types of analyses, which may not be possible, the authors are making very general statements about the general function of morality that are not warranted.

What the previous paragraph also strongly suggests is that the "moral values" items are primarily phrased in the context of relationships (avoiding "doing harm to others," being "faithful and loyal," respecting "hierarchy and authority," etc.). As the authors suggest, some (now-no-longer-so) recent models of morality (notably, Gray's) do stress the dyadic, "it-takes-two-to-tango", nature of morality, but that point has already been made and supported elsewhere, so it seems nearly tautological to phrase the "moral values" in terms of how you should treat others, and then discover that these items are rated as more "important" in social situations -- which could very much mean "relevant." So if moral values are defined in terms in how we treat other people, then it seems hardly surprising that people rate them as more relevant in the presence of others. If the authors were serious about this question, it would be important to create items that do not depend (in their phrasing) on the presence of others, to see if they indeed would seem more important in the presence of others. If it is unavoidable that moral importance items pertain to relationships with other, then again there is something quasi-tautological in the effects reported.

There are also some anomalies that warrant discussion if the authors really want to take these results seriously. Whatever rationale would make them argue that being around parents and best friends raises the importance of morality (scores between 4 and 6), as echoed by the argument on "parochialism" on p.5, why would "child" (2.43) be closer to "stranger" (1.94) and "acquaintance" (1.98)? That does not make any sense from an evolutionary standpoint. That doesn't mean it's wrong, but it cannot be ignored. There's plenty of research suggesting that priming children heightens moral concerns. If this doesn't apply here, this would clarify what the authors are exploring here.

Adding the experimental study was a nice little touch to address the obvious correlational gripe, but it's not a very compelling study, and ends up raising more concerns than it rules out. Different population (US College students, mostly male, age 19, vs. Euro adults, mostly female, M age 36), for starters. That is

not everything, but matters more in this context than in others, and we can't just assume it has no impact. Starting with Carol Gilligan, if we choose to acknowledge that there exist decades of work on this topic prior to this paper, some scholars have argued that women may have been socialized to a morality that is more relational, and would indeed be primed in relations (and according to that body of work, would be erroneously coded as Stage 3 by chauvinistic Kohlbergians). So genders, and all other aspects of socialization, could matter here. There is also the work showing that individuals from interdependent cultures are more sensitive to the "eye" signs, suggesting once more that we cannot be cavalier in this context and assume universality of these effects. This all to say that these things have to be taken into account in this context.

Second, having the experimenter in the room or not may or may not be related to what the authors measure when they asked people who is with them, but that seems a rather loose connection and I am very unsure they can claim that they are manipulating the same construct that is measured in the questionnaire. Upon closer reading, I don't think it's the same DV either (it seems they used the Moral Foundations Questionnaire here), making this dataset even more irrelevant.

Third, it is rather mysterious how the manipulation is performed. It may be perfectly fine but there are many ways to get it wrong, and while we are less careful about these things than we used to be, they still matter. Are participants alone in a cubicle/room or with other participants? When does the experimenter leave/return in the control condition? Do participants expect the experimenter to walk in any minute, or are they guaranteed somehow that they are on their own for while? And what is the actual randomization scheme? Is a coin flipped every time a participant arrives? If we take seriously that just someone in a room has such a strong effect on private answers, we need to be much more meticulous on the precise manipulation of these social factors, and not just describe that the participant "could see" the experimenter, without describing what contact the control participants had (did they take instructions from a rape recording, or did they too interact/see the experimenter, but according to a different schedule?).

Next, I'd like to see the same results without all the controls also (age, gender, non-moral evaluations). While the authors do well to control, I worry given the effect ($p=.027$) that it may be quite sensitive to such choices, so I think we should see both. Relatedly, running only 60 participants for a two-group design is a daring bet... As the authors know, for 80% power, they assumed a d over .70, which would be HUGE for the effect on a morality questionnaire of simply having the experimenter sitting in the room, or more correctly, exiting for part of it. If that is the effect they indeed expected, it warrants headlines much more than the more tepid results of the smartphone study! The fact that experimenters can affect, with a d of .70 or more, simple attitude answers given privately, just by stepping in or outside of the room would mean that this is likely to have affected much of the social psychological research in the last half-century, where experimenters routinely come in and out of rooms as they give attitude questionnaires to participants.

When the authors ran that study with 60 participants, were they going to report it if it wasn't significant? Or what made them confident that this was the appropriate size study? There is a blurb on

p.8 ("might increase power sufficiently") but it is not sufficient or appropriate given the huge leap of faith of running only 60 participants.

~~~~~

Raised eyebrows: Two sloppy details that made me worried there might be more unseen issues with the carefulness of the presentation...

1. There are 10 categories of social others in the questionnaire (see p.13), but there are then 13 categories in Figure 1b... Was there an "other" category that lead to the inclusion of "client," "stranger," or "best friend"? Where do these new categories come from? At the very least it does not raise my confidence in the data.

2. There is a similar surprising lack of precision in the supplementary materials, where the same study is sometimes called 1B, sometimes 2, while another is sometimes 2, sometimes 3. This does not give the impression to have been very carefully crosschecked, assuming that there are not additional studies that are not reported.

~~~~~

In summary, I think the authors discovered an interesting pattern in their data that warrants further exploration and theorizing. I don't think however that the paper comes anywhere close to making the type of contribution it purports to make.

Reviewer #1 (Remarks to the Author):

This paper is interesting and important, and I warmly recommend publication. It's the kind of study we need more of.

Thank you!

Figure 1A makes the effect look pretty modest, a few point difference in “importance” on a 100 point scale. Is that a fair characterization?

Yes, overall our effects, while reliable, are small in absolute terms (d s ~ .15 and .2)—an observation we now reference this in our results (p. 10) and power analyses (see Methods, p. 26 and 27-8). However, these effect sizes are far from trivial in relative terms. We now outline in SOM 1.1 several ways of understanding these results in the context of research on other known shifts in moral evaluation. For example, the estimated 4-point shift in the importance of binding values when one is with an acquaintance versus best friend represents about 27% of the median range of people's fluctuations in moral importance ratings. Moreover, the size of these effects is notable when benchmarked against well-established individual differences in moral importance. For example, research suggests that, as people get older, the importance they ascribe binding values increases approximately .19 of a standard deviation per decade. We find that being in the presence of a best friend versus acquaintance is associated with an approximate increase of about .09 SD. In other words, being in the presence of an acquaintance versus best friend is equivalent to a shift of about half a decade in age. We hope this addition gives readers a better sense of how our effects relate to other known effects in the morality literature.

There is a question about how findings are framed. From the abstract, one gets the impression that the hypothesis whether the “purpose” of morality is social regulation, a question we are told has not been “directly tested” before. I don't think the current study addresses this.

We apologize for our lack of clarity here. Our original intention had been to set the stage for our theoretical assumptions, not to suggest that we had proven that the purpose of morality was for social regulation. Nevertheless, we agree that in our original paper we didn't sufficiently differentiate these claims. In the revised manuscript, while we preserve our claim about the social nature of morality to help readers understand our starting assumptions (first line of the abstract), we now focus our General Discussion on the implications of the results for moral behavior and moral inconsistency (see General Discussion, pp. 21-23).

p. 3: “the emerging view of moral values likens them to rules.” I'm a rules guy myself, but this is contentious.

We have removed this statement.

p. 3. I'm also a fan of watching eyes, but I believe there our grounds for some reservation in the literature.

We have removed reference to this work and now use other research to support the possibility that morals may change based on social context (pp. 5-6).

p. 4. Last sentence of first para, "support." This sentence also strikes me as outrunning what's been shown. The next paragraph begins with a more careful statement, which I take to be the "official" finding.

We have omitted this claim.

p. 5. I more usually see the MFT cashed in terms of binding/individualizing rather than communal/universal.

We have changed all language to "binding/individualizing."

p. 5. Big gender skew for the sample. No gender differences?

We control for gender in all studies to account for any possible effects of gender; also, moral importance does not differ by gender ($p = .37$).

p. 6. I wasn't completely clear on the "positivity effect." Bit more discussion of what was found?

We apologize for the lack of clarity. What we meant when we said positivity effect is the possibility that the effects could be explained by a general tendency for happy participants to answer high on the scale of all questions across the board. If this were the case, then the effect should disappear when one controls for participants' current affective states. As we now explain in the results (p. 10), and more fully in the Supplementary Materials (SOM 1.2), this does not appear to be the case, thus helping to rule out this alternative explanation.

p. 8. First paragraph of the discussion also seems to me to encourage the entangling of the demonstrated association with the speculative purpose I noted with respect to the abstract.

The language has been changed to reflect this point (p. 21).

p. 8. On the novelty of the result. It certainly seems to me novel in terms of directly addressing the paper's official hypothesis (sociality > importance). But it might be (very) worth noting that the relation between sociality and morality has been the object of extensive study.

We have referenced this work in the Introduction (pp. 4-6) and integrate it with other findings in the moral domain in the General Discussion (pp. 21-23).

p. 9. I think it would be worth further underscoring the uncertainties of self-report.

We have now conducted a new laboratory study (Study 2), one of whose primary aims was, by using the Moral Foundations Questionnaire as opposed to single items as the measure of moral importance, to address the potential concern of low reliability with single item scales (see Study 2 Introduction, p. 12). Furthermore, the randomized nature of Study 2 suggests that self-report or experimenter demand effects would not explain the results, as participants expressed no indication that they understood the study design in the experiment debriefing. Finally, we now include a paragraph in the General Discussion (p. 24) reviewing the weaknesses of self-report and calling for future research that extends these observations to more implicit measures.

P. 9. “called this trait-based morality into question.” Here’s another place where controversy ought be noted. Indeed, the controversy was one of the main issues that helped empirically informed, interdisciplinary moral psychology take off, starting around 2000 or so. Relatedly, the choice of citations for this struck me as unusual, in particular by not including any of the classic, widely cited, discussions that are central to the debate, such as Harman, Doris, and Ross and Nisbett. Again, general readers wishing to oriented with respect to this issue would benefit from being pointed to these treatments.

Thank you for these references; they are very important and relevant! We have included these in the General Discussion (p. 23).

p. 10. “morality is social” I’m repeating myself here, but this does not follow directly from people assigning more importance to moral values in social settings.

We apologize for the confusion and have removed this and similar claims from the paper.

Reviewer #2 (Remarks to the Author):

This is an interesting finding, and the unique, real-world methodology deserves praise.

Thank you!

The authors control for age, gender, and time effects, but they do not account for individual differences that are known to be correlated with endorsement of moral foundations, particularly political views.

We take this point seriously; indeed it was a primary focus of Studies 2 and 3, both of which controlled for political orientation (see also Study 1 Discussion, pp. 11-12). We found that controlling for this factor did not impact the results in either investigation. We thank the reviewer for this critical insight and for the opportunity to strengthen the paper by accounting for it in our follow-up investigations.

Another issue is that there is a bit of a mismatch between theory and empirics, I think.

The concern that we overstated our claims was shared by all the reviewers and we have taken care to rectify this issue. In particular, we have removed language suggesting that our results prove that morality is social (though we preserve this language in the abstract to make clear our starting theoretic assumptions). In the General Discussion, we have moved away from making general claims about the nature or purpose of morality and have instead focused on the implications for our observations for moral theory regarding moral tradeoffs and moral malleability. We hope you will find that the language of the paper now more accurately reflects the nature of the findings (pp. 21-23).

-p.6: "Conducting the same analysis without controlling for mood increased the strength of the effect." Yes, but only nominally. The difference here is between $\beta = .089$ and $\beta = .092$. Two effect sizes do not get much more similar than that. This is overstated.

Our main intention here was not to show that the effect *increases* when controlling for mood, but merely that it doesn't *decrease*. We apologize for the poor choice of phrasing. Regardless, we have removed this statement from the text.

-I find it odd that the authors use the terms "universal" and "communal". To harmonize with prior research, I would recommend using the usual terms "individualizing" and "binding".

Done!

-It would be helpful if the authors reported descriptive statistics for each condition in the lab study. With only the regression coefficients presented, it is a bit difficult to interpret what is happening.

We have done so in the newly added Study 2 (see p. 13).

Reviewer #3 (Remarks to the Author):

The authors present an interesting if early finding, that the scores on their "moral values" scale are significantly higher when measured in the presence of others than when not. This wrinkle in their data, perhaps predicted, is intriguing and warrants further inquiry.

Thank you for the encouraging words! We have conducted two new studies, including a large-scale preregistered study, to better understand the effect. The result, we believe, is a manuscript that is considerably sharper and stronger than the original submission. We thank you for your comments and encouragement, which played a large role in the revision.

[The authors' measure] blurs measures of moral motivation, of the self-importance of moral identity a la Aquino and Reed, of as the authors also call it, "moral values" or "moral evaluation." So there is not much to be gained theoretically about the measure of moral self-importance from this measure or this implementation.

We appreciate you calling attention to the lack of clarity in our terms and concepts. In an effort to rectify this issue, we have done several things. First, we clearly define our general measure of moral importance as “importance afforded to [moral] values” in the Introduction (p. 5), and we have eliminated all use of extraneous terms like moral evaluation, moral judgment, and moral identity from the paper. The notion of moral importance is taken directly from original research on values by Schwartz and colleagues. Second, and most important, we now supplement the measure of moral importance from Study 1 with two additional measures of moral importance in Studies 2 and 3. In Study 2, we use the Moral Foundations Questionnaire, which has been widely used to capture the importance people afford moral values. In Study 3, to capture as purely as possible the precise construct we are attempting to describe, we ask participants in direct terms the importance they afford different “decontextualized” moral values (pp. 15-16), as well as complementing this measure with the identical translated measure from Study 1. Moreover, we have added a new paragraph in the General Discussion that explicitly references the links between this study and the work by Aquino & Reed (pp. 22-23). In this way we hope we have adequately addressed the reviewer’s concern about conceptual ambiguity.

The one tradition the authors tip their hat to is Graham and Haidt's five foundations, though I believe they rename Binding/Individualizing into Communal/Universal, presumably because that fits better the current model. This renaming does not seem necessary or warranted, unless the authors claim to be talking about something else.

We agree that this name change was unhelpful and have changed the names back to their original.

The very words chosen, and the fact that they are most relevant in the context of specific relationship, would contribute to higher ratings in social settings, but making it difficult to want to draw big conceptual dimensions out of these 5 specific items.

As we now state in the Study 1 Discussion (pp. 11-12), we agree that the precise wording of the questions from Study 1 might be seen as “begging the question” about the values that would be construed as important in the context of different relationships. This observation is a primary impetus for the designs of Studies 2 and 3, which incorporate different measurements of moral importance in order to provide convergent validity to this finding. We hope that the additional data provided by these studies helps to clarify this issue.

It seems nearly tautological to phrase the "moral values" in terms of how you should treat others, and then discover that these items are rated as more "important" in social situations. If the authors were serious about this question, it would be important to create items that do not depend (in their phrasing) on the presence of others.

In response to this concern, we have done two things. First, we now focus on the effect on moral importance of the presence of *close* others, which, because they

are not explicitly referenced in the question wording, are less subject to the “tautological” concern as the general presence of others. Second, we deliberately tailored Study 2 and 3 to address this point (which we raise explicitly p. 12). Neither the Moral Foundations Questionnaire in Study 2 nor the “moral values” measure in Study 3 depend in their phrasing on the presence of others, thus helping to avoid the possibility that they are demonstrating a tautological effect.

There are also some anomalies that warrant discussion if the authors really want to take these results seriously. Whatever rationale would make them argue that being around parents and best friends raises the importance of morality (scores between 4 and 6), as echoed by the argument on "parochialism" on p.5, why would "child" (2.43) be closer to "stranger" (1.94) and "acquaintance" (1.98)? That does not make any sense from an evolutionary standpoint.

We agree that our results do not fit perfectly into an evolutionary standpoint given the results from Study 1, which finds that children elicit less increases in moral importance than parents or best friends. For this reason, we have moved beyond claims about the effects of specific relationships and focused on the overall effects of social distance on moral importance, and focus less on the evolutionary explanations for this potential effect. Furthermore, we have explicitly highlighted this issue in the Study 1 Discussion (p. 12). Doubtless a variety of complex social factors will play into the degree to which various moral values are activated in the context of particular others, and while we consider this fine-grained analysis a worthy topic of future research, we restrict ourselves in the current paper to exploring the broader differences in the effect of social closeness on different moral values.

Adding the experimental study was a nice little touch to address the obvious correlational gripe, but it's not a very compelling study, and ends up raising more concerns than it rules out.

We agree with the reviewer’s many concerns about this study and have removed it from the manuscript. Because this study has been eliminated from the paper, we have omitted the remainder of the reviewer’s comments concerning this study since they are no longer relevant. We hope that the inclusion of both a new laboratory experiment and large-scale, preregistered online study have adequately addressed the reviewer’s additional concerns.

Raised eyebrows:

1. There are 10 categories of social others in the questionnaire (see p.13), but there are then 13 categories in Figure 1b... Was there an "other" category that lead to the inclusion of "client," "stranger," or "best friend"? Where do these new categories come from? At the very least it does not raise my confidence in the data.

We apologize for the error and have rectified it in the Methods of the revised manuscript (p. 25)

2. There is a similar surprising lack of precision in the supplementary materials, where the same study is sometimes called 1B, sometimes 2, while another is sometimes 2, sometimes 3.

We have rectified this issue.

REVIEWER COMMENTS

Reviewer #1 (Remarks to the Author):

I have reviewed the letter and revised MS, and am satisfied with the responses, and therefore continue to recommend publication.

One small point, on this bit of text:

"research in the past few decades has questioned whether moral character exists at all"

It's not clear that this was ever anyone's considered view, except maybe Skinner. Harman sometimes overstated things, but he in fact explicitly rejected the claim. The real source of this mischaracterization was defenders of character theory "strawmanning" their opponents, but as has been repeatedly pointed out, the question is not whether character or traits of character "exist," but how they should be characterized. What the skeptic doubts is that moral character has the sort of substantial and pervasive influence on behavior that folk psychology, and some theory, assumes. In short, the effect sizes associated with individual difference variables like traits are smaller than might a priori be expected. Worth keeping the record straight, to help preclude superfluous discussion in the future -- esp, since things could be reworded so easily.

Reviewer #2 (Remarks to the Author):

This paper is considerably improved since the previous draft. My concerns about overstated theory and political affiliation as an alternative explanation for the findings have been addressed.

However, the focus of the paper has shifted since the previous version. The previous draft argued that "moral values were rated as more important overall when people were in the presence of others versus alone". A secondary finding was that this was particularly true for binding moral values over individualizing values. Now, however, the claim is that binding values, but not individualizing values, become more important in the presence of *close* others, specifically. This change has introduced some new issues to the paper.

MAJOR ISSUES

Normally, the new pre-registered study would alleviate my concerns about "hypothesizing after results are known" ("HARKing"), but, looking at the pre-registration, the original finding (presence of others leads to greater overall moral importance) was pre-registered as "Hypothesis 1". This is not mentioned

anywhere in the paper, and the analysis is not reported. This seriously violates the spirit of pre-registration, and it seems like the authors simply put the focus on what "worked" (Hypothesis 3). This is a problem.

This shift in focus led to the change in analytic approach to Study 2. Obviously, splitting the "partner" condition into "close" and "not close" groups is not ideal. It would be much better to directly manipulate social closeness somehow (preferably in a new pre-registered lab study). I didn't expect to be asking for more new data at this point in the review process, but I think the looseness with which the authors treated the Study 3 pre-registration makes this necessary. The paper needs a study that clearly pre-registers the prediction that the importance of the binding values will increase in the presence of close others (and only this prediction), then manipulates social closeness.

Relatedly, the follow-up analysis in Study 2 is very odd. Social closeness should not be treated as a linear predictor (alone vs. distant vs. close cannot be said to "equidistant" from each other in any sense). I do not understand why the authors did not simply run post-hoc tests on their ANOVA. That would be the standard way to probe the interaction effect.

MINOR ISSUES

It is also odd that the authors mention "greed" as related to the Purity foundation (p. 4). I've never heard this before, and I've been doing moral psychology for quite some time.

The x-axis in Figure 2A is not numbered.

It is not true that "because participants were randomly assigned to either the 'alone' or 'partner' condition" that political conservatism cannot explain the results of Study 2 (p. 16). The action in this study is clearly among the participants who felt "close" to their partner. But, of course, people self-select into this group, so the third variable problem remains here. However, the authors did statistically control for political affiliation in this study, which does take care of this problem. That is where the focus should be in the Discussion.

CONCLUSION

Upon reading the abstract to this revised manuscript, I expected this to be a very short review that would move this paper along to publication. But, upon reading the paper, I am very concerned about the authors burying analyses that did not work and completely changing the hypothesis of Study 2 after the fact, resulting in a sub-optimal test of the new hypothesis. Researchers used to get away with this kind of post-hoc theorizing all the time, but times have changed (for the better, in my view), and this simply is not best practice.

Reviewer #3 (Remarks to the Author):

I served as Reviewer #3 in the initial submission and I continue to see a lot of promise in this work, and congratulate the researchers on the terrific work they have done carrying this project forward. And it's definitely come a long way. Study 3 largely replicates Study 1, but with a new sample and new scales, and it's registered, which helps. The authors do a good job pointing out the weaknesses of Study 1 now, and promise us a random experiment to remedy its faults Except that after they do run one with Study 2 (and it's got other issues, lots is confounded between "alone" and "partner" so it would be hard to pinpoint the manipulation even if it did have an impact), they then toss away the random manipulation by doing an internal analysis based on participants' manipulation check.

The biggest concern is with Study 2, which claims to move from a correlational approach to an experiment with random assignments, but then splits the "partner" condition into 2 groups based on participant responses ("close" vs "distant"), thus in effect breaking the random assignment... There's no justification for doing this, it's an obvious major violation of random assignment. It could maybe be done as an internal analysis late in the discussion when speculating about an obtained pattern, but definitely not as the main analysis. As randomized, the mean difference would be something like 4.86 for alone vs 4.96 with a partner, a difference of .10 which is the proper comparison.

****This absolutely needs to be fixed before the paper can be considered further.****

If this difference is significant, we can start discussing what, of the many things are confounded, is likely at work here (the presence of another, the fact that they asked them questions, the fact that they answered some, ... etc.?). If it's NOT significant, it's a much greater concern for the authors. It doesn't mean that the data should not be published, but just that all the concerns attached to the correlational nature of Studies 1 and 3 remain, and that it's much harder to know what to make of all these data.

I also was confused by the analyses of Study 2. On p.14 the design is "a 3 (social context: alone v. close partner v. distant partner) × 2 (value type: binding versus individualizing) mixed-model ANOVA", but in the methods (p.28) it's described as "the primary predictors were value type and social closeness, operationalized both as a manipulated factor (partner versus alone) and partner closeness (close versus distant)," as if they were separate crossed predictors with 1df each. I understand breaking a 2df omnibus into 1df contrast, but that would be (alone vs close) and (alone vs distant). Or perhaps the authors mean that this is two orthogonal contrasts, with one nested in the other $[(-2,1,1)$ and $(0,-1,1)]$? Either way, this analytical model doesn't quite line up with what's in the results of the main text, with the 2df omnibus test.

Unfortunately for the authors, their manipulation check suggests a failed manipulation of closeness. They asked about closeness on a scale ranging from "Not at all close" (1) to "Very close" (5). One hundred and twenty participants (67%) in the "close partner" condition apparently chose "Not at all", while the rest chose "somewhat close" or above... (Not sure if there is a scale point between not at all and somewhat, AND I can't find the distribution of responses, or mean or SD on closeness anywhere). So yes, the manipulation largely failed according to the manipulation check, and all the authors can do is

see if the presence of a stranger (confounded with answering personal questions about oneself, etc.) is different from being alone. This CANNOT be used as a random manipulation of the presence of a close other, which is the whole point of this study. Even the 59 categorized as “close partner” are a stretch since some participants tentatively answered “somewhat close” (Or maybe more? We don't know)...

As a side point, I don't think as a result that Study 2 casts any light on the anomalous findings in Study 1 regarding children and romantic partners, which the authors do acknowledge but then promise to address in Study 2, which Study 2 doesn't.

Study 3 is a nice addition to the package, but mostly is a replication of Study 1, with all the issues that the authors themselves identified (quite thoroughly, if concisely, I must admit on pp. 12-13) when they inventory the shortcomings of Study 1 as correlational study. What's nice is that it extends the work to the MFQ dimensions, which makes it easier for this work to connect with other research.

A detail: The “ribbons” representing the error around the estimates seem to be missing in Fig 2A and 4.

Reviewer #1

I have reviewed the letter and revised MS, and am satisfied with the responses, and therefore continue to recommend publication.

Thank you!

**"research in the past few decades has questioned whether moral character exists at all"
It's not clear that this was ever anyone's considered view.**

Thank you for this important clarification. We have made the appropriate change in the manuscript. Specifically, we said (p. 33):

...recent research on the contextual sensitivity of moral judgment and behavior has called this trait-based approach to morality into question. The current findings suggest one specific source giving rise to within-person variability or "inconsistency" in morality, by suggesting that, far from being immutable, certain moral values—in particular, binding values—may be flexibly applied according to the demands of the situation.

Reviewer #2

This paper is considerably improved since the previous draft.

Thank you!

Looking at the pre-registration, the original finding (presence of others leads to greater overall moral importance) was pre-registered as "Hypothesis 1". This is not mentioned anywhere in the paper, and the analysis is not reported. This is a problem. I am very concerned about the authors burying analyses. Researchers used to get away with this kind of post-hoc theorizing all the time, but times have changed (for the better, in my view), and this simply is not best practice.

We agree with the reviewer that failing to report preregistered hypotheses would be a serious violation of the spirit of preregistration. We apologize for this misunderstanding; we should have made clearer that all preregistered analyses were available in the Supplementary Online Materials (included with our submission). Now, in accordance with the editor's suggestion that "pre-registered analyses [be] conducted and presented in the main text," we have moved *all* preregistered analyses pertaining to this study to the primary document. The reviewer will find the full description of preregistered hypotheses pertaining to Study 3 on pp. 15-16. Specifically, we say:

In accordance with the preregistration, we examined the main effects of social context (being alone versus with others) on the overall importance of moral behavior and values (i.e., collapsing across binding and individualizing values). Results showed no significant effect for moral behavior, $B = -0.52$, $SE = 0.68$, $t(1993) = -0.77$, $p = 0.44$, but a negative effect for moral values, $B = -1.89$, $SE = 0.58$, $t(2718) = -3.25$, $p = 0.001$. This

appears to be driven by the negative relationship between social closeness and individualizing values, possible explanations for which are discussed below.

Next, we tested the overall effect of social closeness (estimated and self-rated) on the overall importance of moral behaviors and values (i.e., collapsing across binding and individualizing values). For estimated closeness, results showed no significant effect on moral values, $B = -0.01$, $SE = 0.01$, $t(2718) = -1.44$, $p = 0.149$, and a marginally negative effect on moral behavior, $B = -0.01$, $SE = 0.01$, $t(2713) = -1.83$, $p = 0.067$. On the other hand, for self-rated closeness, results showed a significant positive relationship between self-rated closeness for both moral behaviors, $B = 0.01$, $SE = 0.01$, $t(2713) = 1.99$, $p = 0.046$, and moral values, $B = 0.02$, $SE = 0.01$, $t(2718) = 3.27$, $p = 0.001$. These mixed results, which are likely due to variations in the relative impact of binding versus individualizing value sets, further support the idea that these different value types are best considered separately (i.e., as an interaction effect), as opposed to as a single category.

In addition, the latest version of the manuscript now includes 3 additional preregistered studies. The first of these, Study 4A, did not yield the expected result (see pp. 23-25). We describe the study in detail in the main text and include all pre-registered hypotheses. As we explain in greater detail in the paper, we report on this experiment not just for transparency, but because we believe this “failed” experiment substantially contributes to understanding the phenomenon described in the paper. This is especially true in conjunction with our “successful” studies 4B and 4C because it suggests boundary conditions in which feelings of social distance may elicit compensatory activation of moral values—a phenomenon which, while beyond the scope of this paper, is a potentially fruitful topic for future research (see p. 34).

Overall, we thank the reviewer for advocating for even greater transparency and hope we can model in this paper the type of research practices necessary for robust and replicable psychological science.

This shift in focus led to the change in analytic approach to Study 2. Obviously, splitting the "partner" condition into "close" and "not close" groups is not ideal.

We agree that this approach was not ideal. It would have been better to simply observe a difference between the “alone” and “partner” conditions without including perceived partner closeness as an additional moderator. As we now highlight in greater detail in the paper, the effect didn’t emerge under this analysis, likely due to the fact that our manipulation of partner closeness was unsuccessful (see p. 19). We hope our revised description of Study 2 (in particular, the study Discussion) adequately addresses the project’s strengths and limitations. As we now write (p. 24):

Despite these promising exploratory results, the failure of the primary experimental manipulation renders all these findings provisional. Because we did not observe this predicted effect, we were forced to rely on a quasi-experimental design in which we examined the effect of being in person among those who said they felt close or distant from their partner. This introduces the possibility that a third variable is simultaneously causing people to rate their partner as close and binding values as

important. Thus it will be important to test these results using an approach that doesn't rely on measured social closeness to corroborate the results.

The paper needs a study that clearly pre-registers the prediction that the importance of the binding values will increase in the presence of close others (and only this prediction), then manipulates social closeness.

We're pleased to report that we now include two studies (Study 4B and 4C) that do indeed find that the importance of binding values increases in the presence of close (vs. distant) others. The preregistrations, which can be found at <http://aspredicted.org/blind.php?x=fr9fd3> and <https://aspredicted.org/blind.php?x=f9796m>, predict that the importance of binding values will be higher when people imagine being in the presence of close other than a distant other. And indeed, we find strong support for this prediction, all $ps < .0001$. Moreover, in exploratory analyses, we show that this effect interacts with value type (binding versus individualizing) such that binding values are on average significantly more affected by relationship type than individualizing ($p < .001$). The full descriptions of the studies can be found on pp. 26-30, with a figure representing these findings on p. 30:

*Figure 5. Binding values gain importance more than individualizing values in the presence of close others. Each panel ($n = 580$ and $n = 752$, respectively) shows the average rated importance of binding versus individualizing values when people are in the imagined presence of another person. In Study 4B, participants imagined being in the presence of a romantic partner or a stranger; in Study 4C participants imagined being in the presence of someone to whom they “feel close” or “do not feel close.” Error bars represent 95% confidence intervals around the mean. * $p < .001$*

The follow-up analysis in Study 2 is very odd. Social closeness should not be treated as a linear predictor (alone vs. distant vs. close cannot be said to "equidistant" from each other

in any sense). I do not understand why the authors did not simply run post-hoc tests on their ANOVA. That would be the standard way to probe the interaction effect.

We appreciate this point and are happy to include the results of the pairwise comparison. We are also including the analysis with closeness as a linear predictor in a footnote. We believe this analysis is important to include because we treat closeness as a linear predictor in Studies 1 and 2 and think that some readers would be interested to see effects in Study 3 analyzed in a similar manner. Specifically, we say (p. 20):

A test of this hypothesis using closeness as a linear predictor with "alone" coded as 0 (as in Studies 1 and 2) yielded a similar pattern of effects, $B = -0.06$, $SE = 0.03$, $t(346) = -1.96$, $p = 0.051$.

And:

Furthermore, a post hoc comparison of the importance of binding values among participants who were alone versus those who rated their partner as close showed a significant difference, $t(107) = 2.09$, $p = .039$; supporting the claim that people who were alone rated binding values less important than those who rated their partner as close.

It is also odd that the authors mention "greed" as related to the Purity foundation (p. 4). I've never heard this before, and I've been doing moral psychology for quite some time.

This was taken from the paper by Graham, Haidt, & Nosek (2009; p. 1031, 2nd paragraph). We have now cited the reference. Nevertheless, if the reviewer still feels it is out of place, we are happy to remove it.

The x-axis in Figure 2A is not numbered.

We have rectified the error (p. 11).

It is not true that "because participants were randomly assigned to either the 'alone' or 'partner' condition" that political conservatism cannot explain the results of Study 2 (p. 16). The action in this study is clearly among the participants who felt "close" to their partner. But, of course, people self-select into this group, so the third variable problem remains here. However, the authors did statistically control for political affiliation in this study, which does take care of this problem. That is where the focus should be in the Discussion.

We've changed the Discussion to more accurately reflect this point. First, we have removed the point about participants being randomly assigned to the alone or the partner condition, which we agree was misleading. Moreover, we now say (p. 22):

The fact that these results held controlling for demographic variables, political affiliation, current affective state, non-moral evaluation, and social desirability suggests these variables are not responsible for the observed pattern of effects.

Reviewer #3:

I continue to see a lot of promise in this work, and congratulate the researchers on the terrific work they have done carrying this project forward.

Thank you!

The biggest concern is with Study 2, which claims to move from a correlational approach to an experiment with random assignments, but then splits the “partner” condition into 2 groups based on participant responses ("close" vs "distant"), thus in effect breaking the random assignment

Reviewer 2 identified many of the same flaws in Study 2, and we recognize that the way we presented Study 2 (in particular, inadequately addressing the concerns that arise when including a manipulation check in the analysis) inferred a causal relationship when it was not warranted. We have endeavored to re-write Study 2 to more accurately reflect the results (see pp. 19-23, with a particular emphasis on the Discussion, p. 23).

In addition, we have executed three additional preregistered experiments (Studies 4A-C) in order to more conclusively demonstrate the predicted effect. We hope that, when taken together, these studies present a convincing pattern of effects.

I also was confused by the analyses of Study 2. On p.14 the design is “a 3 (social context: alone v. close partner v. distant partner) × 2 (value type: binding versus individualizing) mixed-model ANOVA”, but in the methods (p. 28) it’s described as “the primary predictors were value type and social closeness, operationalized both as a manipulated factor (partner versus alone) and partner closeness (close versus distant),” as if they were separate crossed predictors with 1df each.

We apologize for the confusion here. What we did was take the people in the “partner” condition and divide them into a “close” versus “distant” partner group. Thus the “social context” variable consisted of three groups: alone, distant partner, and close partner. This 3-level factor was then crossed with value type (binding versus individualizing) to create a 3 x 2 mixed-model ANOVA. We have clarified this in the text (p. 19):

Because we did not obtain the results we had predicted, we conducted a series of exploratory analyses to understand why. We first examined the frequency distribution of the closeness measure. Results showed that only 59 participants (32%) felt even “somewhat close” to their partner, suggesting a failure of our manipulation of social closeness. Given that our predictions suggested only people who felt close to their interaction partner would show an increase in the importance of binding values, we reasoned that the low number of participants in the former category might explain the absence of the predicted effect.

To test this possibility, we re-analyzed the results with a focus on the people who felt close to their partner. This approach undermines the random assignment strategy of

the experimental design, and so we note that these effects cannot be interpreted as providing insight into causality. Nevertheless, this exploratory analysis can provide a better understanding of our obtained results. We split the participants in the “partner” condition into “distant” partner and “close partner” categories according to whether they indicated they felt at least “somewhat close” to their partner, and treated these as separate conditions. We then conducted a 3 (social context: alone vs. distant partner vs. close partner) × 2 (value type: binding vs. individualizing) mixed-model ANOVA with moral importance as the dependent variable, controlling for all incidental variables. As seen in Figure 3, the analysis revealed a significant interaction between value type and social context, $F(2, 345) = 4.02, p = .019$

Unfortunately for the authors, their manipulation check suggests a failed manipulation of closeness. They asked about closeness on a scale ranging from “Not at all close” (1) to “Very close” (5). One hundred and twenty participants (67%) in the “close partner” condition apparently chose “Not at all”, while the rest chose “somewhat close” or above...

You're right, and we thank you for helping us to better understand why we didn't find our anticipated effect. We have given this much thought, and it helped us to eventually design an effective experimental manipulation of closeness in Studies 4B and 4C.

I can't find the distribution of responses, or mean or SD on closeness anywhere

We have now included fuller descriptive statistics of this measure, p. 19:

The mean level of partner closeness was 2.11, $SD = .98$.

All the authors can do is see if the presence of a stranger (confounded with answering personal questions about oneself, etc.) is different from being alone. This CANNOT be used as a random manipulation of the presence of a close other

We agree. We now highlight this to a greater extent in the discussion. Specifically, we say (p. 22):

Despite these promising exploratory results, the failure of the primary experimental manipulation renders all these findings provisional. Because we did not observe this predicted effect, we were forced to rely on a quasi-experimental design in which we examined the effect of being in person among those who said they felt close or distant from their partner. This introduces the possibility that a third variable is simultaneously causing people to rate their partner as close and binding values as important. Thus it will be important to test these results using an approach that doesn't rely on measured social closeness to corroborate the results.

I don't think as a result that Study 2 casts any light on the anomalous findings in Study 1 regarding children and romantic partners, which the authors do acknowledge but then promise to address in Study 2, which Study 2 doesn't.

We apologize for the confusion. After some consideration, we have decided to remove these analyses from the manuscript for reasons of clarity/readability. However, we wanted to make sure we thoroughly address the reviewer’s concern. To do this, we ran a series of analyses in which we compared the means of people in these social contexts using a series of t-tests. Results from showed that people in presence of children or romantic partners judged binding values more important than those in presence of strangers or acquaintances in every pairwise comparison combination, all $ps < .004$, with the exception of a marginally significant difference between stranger and romantic partner, $p = .053$, and a nonsignificant difference between those with a romantic partner and those with an acquaintance, $p = .27$.

We hope this helps allay the concern that binding values are not rated as more important when people in the context of these particular close relationships. If the reviewer prefers that we include this analysis in the manuscript itself, we are happy to do so.

A detail: The “ribbons” representing the error around the estimates seem to be missing in Fig 2A and 4.

We apologize for this! In re-submitting the manuscript we realized that the PDF renderer was omitting ribbons for *all* the graphs we’d created! We’ve now (hopefully) rectified this error.

REVIEWERS' COMMENTS

Reviewer #2 (Remarks to the Author):

I am happy to now be writing the review that I expected to be writing last time around: the authors have addressed all of my concerns, and I am pleased to recommend publication of this very interesting paper. In particular, the authors have greatly improved the transparency of their studies, acknowledged the shortcomings of their methods, and present new, pre-registered, experimental evidence for their hypothesis. I commend them for all of this.

Reviewer Comments.

Reviewer #2 (Remarks to the Author):

I am happy to now be writing the review that I expected to be writing last time around: the authors have addressed all of my concerns, and I am pleased to recommend publication of this very interesting paper. In particular, the authors have greatly improved the transparency of their studies, acknowledged the shortcomings of their methods, and present new, pre-registered, experimental evidence for their hypothesis. I commend them for all of this.

Thank you! We appreciate your help in this process!